# Better Learning-Augmented Spanning Tree Algorithms via Metric Forest Completion

**Nate Veldt[1], Thomas Stanley[1], Benjamin W. Priest[2], Trevor Steil[2], Keita Iwabuchi[2],**
**T.S. Jayram[2], Grace J. Li[2], Geoffrey Sanders[2]**
[1]Texas A&M University, [2]Lawrence Livermore National Laboratory
`{nveldt,thomas.stanley}@tamu.edu`
`{priest2,steil1,iwabuchi1,thathachar1,li85,sanders29}@llnl.gov`

## Abstract

We present improved learning-augmented algorithms for finding an approximate minimum spanning tree (MST) for points in an arbitrary metric space. Our work follows a recent framework called metric forest completion (MFC), where the learned input is a forest that must be given additional edges to form a full spanning tree. Veldt et al. (2025) showed that optimally completing the forest takes $\Omega(n^2)$ time, but designed a 2.62-approximation for MFC with subquadratic complexity. The same method is a $(2\gamma + 1)$-approximation for the original MST problem, where $\gamma \geq 1$ is a quality parameter for the initial forest. We introduce a generalized method that interpolates between this prior algorithm and an optimal $\Omega(n^2)$-time MFC algorithm. Our approach considers only edges incident to a growing number of strategically chosen "representative" points. One corollary of our analysis is to improve the approximation factor of the previous algorithm from 2.62 for MFC and $(2\gamma + 1)$ for metric MST to 2 and $2\gamma$ respectively. We prove this is tight for worst-case instances, but we still obtain better instance-specific approximations using our generalized method. We complement our theoretical results with a thorough experimental evaluation.

## 1 Introduction

Finding a minimum spanning tree (MST) of a graph is a fundamental computational primitive with applications to hierarchical clustering (Gower & Ross, 1969; Gagolewski et al., 2025; La Grassa et al., 2022), network design (Loberman & Weinberger, 1957), feature selection (Labbé et al., 2023), and even comparing brain networks (Stam et al., 2014). The *metric* MST problem is a special case where the input is a set of $n$ points, and edge weights are defined by distances between points. A conceptually simple algorithm for this case is to compute all $O(n^2)$ distances explicitly and then apply a classical greedy algorithm. For Euclidean metrics, there exist more sophisticated algorithms that can find an optimal or at least approximate minimum spanning tree in $o(n^2)$ time (Agarwal et al., 1990; Shamos & Hoey, 1975; Vaidya, 1988; Arya & Mount, 2016). For general metric spaces, however, one must know $\Omega(n^2)$ edges to compute even an approximate solution (Indyk, 1999). This fact constitutes a fundamental challenge for designing algorithms that scale to massive modern datasets, apply to general distance functions, and come with provable guarantees.

Motivated by the above challenge, our previous work (Veldt et al., 2025) recently addressed the metric MST problem from the perspective of *learning-augmented* algorithms (Mitzenmacher & Vassilvitskii, 2022). The learning-augmented model assumes access to a prediction for some problem, often produced by a machine learning heuristic, that comes with no theoretical guarantees but may still be useful in practice. The goal is to design an algorithm that can obtain better than worst-case guarantees using the prediction. The performance of the algorithm is typically captured by some parameter measuring the error of the prediction. The prediction, performance measure, and error parameter vary depending on the context. Some prior work focuses on better-than-worst-case runtimes or query complexities, including for binary search (Mitzenmacher & Vassilvitskii, 2022; Dinitz et al., 2024), maximum flow (Polak & Zub, 2024; Davies et al., 2024; 2023), and incremental approximate shortest paths (McCauley et al., 2025). Other works focus on improving competitive ratios for online algorithms, including for ski rental (Mitzenmacher & Vassilvitskii, 2022; Shin

et al., 2023), scheduling (Benomar & Perchet, 2024), and online knapsack problems (Lechowicz et al., 2024). In other settings, the goal is to improve approximation ratios for hard combinatorial problems, e.g., clustering problems (Braverman et al., 2025; Ergun et al., 2022; Nguyen et al., 2023; Huang et al., 2025) or maximum independent set (Braverman et al., 2024).

For the metric MST problem, our recent work (Veldt et al., 2025) introduced a learning-augmented setting where the $n$ points are partitioned into components and each component is associated with a tree on its points. This input is called the initial forest, and can be viewed as a prediction for the forest that would be obtained by running several iterations of a classical algorithm such as Kruskal's. *Metric forest completion* (MFC) is then the task of finding a minimum-weight spanning tree that contains the initial forest as a subgraph. The quality of an initial forest is captured by a parameter $\gamma \geq 1$, where $\gamma = 1$ if the initial forest is contained in some optimal MST. We previously proved that optimally solving MFC takes $\Omega(n^2)$ time, but gave a 2.62-approximation algorithm whose runtime depends on the number of components $t$, and has subquadratic complexity if $t = o(n)$ (Veldt et al., 2025). The same method is a learning-augmented algorithm for metric MST with an approximation factor of $(2\gamma + 1)$. The algorithm identifies a single *representative* node for each component in the initial forest, and only considers edges incident to one or two representatives. Implementations of the algorithm produced nearly optimal spanning trees while being orders of magnitude faster than the naive $\Omega(n^2)$ algorithm for metric MST. This is true even after factoring in the time to compute an initial forest. The in-practice approximation ratios also far exceeded the theoretical bounds of $(2\gamma + 1)$ (for the original MST problem) and 2.62 (for the MFC step) on all instances.

**Our contributions: generalized algorithm and tighter bounds.** While our prior work already demonstrates the theoretical and practical benefits of the MFC framework, several open questions remain. Is the large gap between theoretical bounds and in-practice approximation ratios due mainly to the specific datasets considered? Are there pathological examples where the previous approximation guarantees are tight? In the other direction, can we tighten the analysis to improve the worst-case approximation guarantees? Also, can we prove better instance-specific approximations?

We introduce and analyze a generalized approximation algorithm for MFC that provides a way to address all of these questions. This algorithm starts with a budget for the number of points in the dataset that can be labeled as representatives. It then finds the best way to complete the initial forest by only adding edges incident to one or two representatives. Choosing one representative per component corresponds to applying our prior approximation algorithm (Veldt et al., 2025). Letting all points be representatives leads to an optimal (but $\Omega(n^2)$-time) algorithm. Our new approach interpolates between these extremes, and for reasonable-sized budgets provides a way to significantly improve on the prior algorithm with only minor increase in runtime. We derive new instance-specific bounds on the approximation factor for this generalized approach, given in terms of an easy-to-compute cost function associated with a set of representatives. As an important corollary of our theoretical results, we prove that when there is only one arbitrary representative per component, the algorithm is a 2-approximation for MFC and a $2\gamma$-approximation for metric MST. This immediately improves on the approximation factors of 2.62 and $(2\gamma + 1)$. Furthermore, our analysis is both simpler and more general. We also prove by construction that these guarantees are tight in the worst case.

As a technical contribution of independent interest, we show that choosing the best set of representatives for our algorithm amounts to a new generalization of the $k$-center clustering problem. For this generalization, we have multiple instances of points to cluster, but the budget $k$ on the number of cluster centers is shared across instances. We design a 2-approximation for this shared-budget multi-instance $k$-center problem by combining a classical algorithm for $k$-center (Gonzalez, 1985) with a dynamic programming approach for allocating the shared budget across different instances.

As a final contribution, we test an implementation of our new algorithm on a range of real-world datasets with varying distance metrics. We find that increasing the number of representatives even slightly leads to significant improvements in spanning tree quality with only a small increase in runtime, and that our dynamic programming approach performs especially well. Furthermore, our instance-specific approximation guarantees are easy to compute and serve as a very good proxy for the true approximation factor, which is impractical to compute exactly.

## 2 PRELIMINARIES AND RELATED WORK

For $m \in \mathbb{N}$, let $[m] = \{1, 2, \ldots, m\}$. For an undirected graph $G = (V, E)$ and edge weight function $w: E \to \mathbb{R}$, a minimum spanning tree (MST) for $G$ with respect to $w$ is a tree $T = (V, E_T)$ where $E_T \subseteq E$ and the total weight of edges $w(E_T) = \sum_{e \in E_T} w(e)$ is minimized. Optimal greedy algorithms for this problem have been known for nearly a century (Borůvka, 1926; Kruskal, 1956; Prim, 1957). For example, Kruskal's algorithm starts with all nodes in singleton components, and at each step adds a minimum weight edge that connects two disjoint components. Borůvka's algorithm is similar, but adds the minimum weight edge adjacent to *each* component every round.

**The metric MST problem.** Let $(\mathcal{X}, d)$ be a finite metric space defined by a set of points $\mathcal{X} = \{x_1, x_2, \ldots, x_n\}$ and a distance function $d: \mathcal{X} \times \mathcal{X} \to \mathbb{R}^+$. This input implicitly defines a complete graph $G_{\mathcal{X}} = (\mathcal{X}, E_{\mathcal{X}})$ with an edge function $w_{\mathcal{X}}$ that is equivalent to the distance function $d$. We let $(u, v)$ denote the edge in $G_{\mathcal{X}}$ defined by points $(x_u, x_v)$, with weight $w_{\mathcal{X}}(u, v) = d(x_u, x_v)$. For two sets $X, Y \subseteq \mathcal{X}$, define $d(X, Y) = \min_{x \in X, y \in Y} d(x, y)$. We extend $w_{\mathcal{X}}$ to a weight function on an edge set $F \subseteq E_{\mathcal{X}}$ by defining $w_{\mathcal{X}}(F) = \sum_{(u,v) \in F} w_{\mathcal{X}}(u, v)$. The metric MST problem is the task of finding a minimum spanning tree of $G_{\mathcal{X}}$ with respect to $w_{\mathcal{X}}$.

A conceptually simple approach for metric MST is to query all $O(n^2)$ distances and apply a classical algorithm to the resulting complete graph. Another approach that still takes $\Omega(n^2)$ time for general metric spaces but avoids querying all distances is an *implicit* implementation of a classical method, which only queries distances as needed (Agarwal et al., 1990; Callahan & Kosaraju, 1993). In more detail, an implicit implementation of Kruskal's or Borůvka's algorithm starts with all $n$ points as singleton components. At every step of the algorithm, for each pair of components $A$ and $B$, the algorithm finds a pair of points $(a, b) \in A \times B$ with minimum distance. This is known as the bichromatic closest pair problem (BCP) for $A$ and $B$. An implicit implementation of Kruskal's algorithm would then add the minimum weight edge from among all the BCP problems. An implicit implementation of Borůvka's algorithm would add one edge for each component.

**The initial forest for learning-augmented MST.** When applying Kruskal's or Borůvka's algorithm implicitly to $\mathcal{X}$, terminating the algorithm early would produce a forest of disconnected components (see Figure 1a). Inspired by this observation, we recently introduced a learning-augmented framework for metric MST where the input can be viewed as a heuristic prediction for the forest that would be produced by terminating a classical algorithm early (Veldt et al., 2025). Formally, an *initial forest* $G_t = (\mathcal{X}, E_t)$ for $(\mathcal{X}, d)$ is defined by a partitioning $\mathcal{P} = \{P_1, P_2, \ldots, P_t\}$ of $\mathcal{X}$ and a partition spanning tree $T_i = (P_i, E_{T_i})$ for each $i \in [t]$ such that $E_t = \cup_{i=1}^t E_{T_i}$. See Figure 1c. Let $P(x)$ denote the partition $x \in \mathcal{X}$ belongs to in $\mathcal{P}$. We say $T_i$ is the $i$th component of $G_t$.

Terminating an exact algorithm early to find an initial forest is prohibitively expensive if one wants to avoid quadratic complexity. One alternative is to run a fast clustering heuristic (e.g., the simple 2-approximation for $k$-center Gonzalez (1985)) to partition $\mathcal{X}$, and then recursively find an approximate or exact MST for each partition. Another approach is to compute an approximate $k$-nearest neighbors graph for $G_{\mathcal{X}}$ and then find a spanning forest of it. These and other similar strategies have already been used in prior work to develop fast heuristics (without approximation guarantees) for *Euclidean* MSTs (Almansoori et al., 2024; Chen, 2013; Zhong et al., 2015; Jothi et al., 2018). One contribution of our prior work (Veldt et al., 2025) was to formalize the notion of an initial forest and introduce a way to measure its quality. To define this measure, let $\mathcal{T}_{\mathcal{X}}$ denote the set of MSTs of $G_{\mathcal{X}}$. For a tree $T \in \mathcal{T}_{\mathcal{X}}$, let $T(\mathcal{P}) = \{(u, v) \in T : P(u) = P(v)\}$ be the set of edges from $T$ whose endpoints are from the same partition of $\mathcal{P}$. The $\gamma$-overlap of $\mathcal{P}$ is defined to be

$$\gamma(\mathcal{P}) = \frac{w_{\mathcal{X}}(E_t)}{\max_{T \in \mathcal{T}_{\mathcal{X}}} w_{\mathcal{X}}(T(\mathcal{P}))}.$$

In other words, $\gamma(\mathcal{P})$ captures the weight of edges that the initial forest has in $\mathcal{P}$, divided by the weight of edges that an optimal MST places inside components. Lower values of $\gamma$ are better, as they indicate that the initial forest overlaps well with some optimal solution. One can use the minimizing property of MSTs to show that $\gamma(\mathcal{P}) \geq 1$, with equality exactly when $G_t$ is contained inside some optimal MST. When $\mathcal{P}$ is clear from context, we will simply write $\gamma = \gamma(\mathcal{P})$.

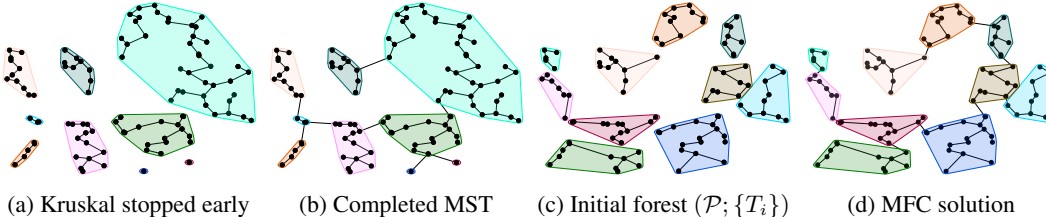

(a) Kruskal stopped early    (b) Completed MST    (c) Initial forest $(\mathcal{P}; \{T_i\})$    (d) MFC solution

Figure 1: (a) The forest obtained by terminating Kruskal's algorithm early for a set of 100 points. (b) Running Kruskal's algorithm to the end leads to a full MST. (c) The initial forest can be viewed as a heuristic prediction for the forest in (a). For this example, $\gamma(\mathcal{P}) \approx 1.06$. (d) Solving metric forest completion problem produces a full spanning tree that approximates the true MST.

**Metric forest completion.** Given an initial forest, *Metric Forest Completion* (MFC) is the task of finding a minimum weight spanning tree that contains $E_t$ as a subgraph. Formally:

$$
\begin{aligned}
\text{minimize} \quad & w_{\mathcal{X}}(E_T) \\
\text{subject to} \quad & T = (\mathcal{X}, E_T) \text{ is a spanning tree for } G_{\mathcal{X}} \\
& E_t \subseteq E_T.
\end{aligned}
\tag{1}
$$

This is equivalent to finding a minimum weight set of edges $M \subseteq E_{\mathcal{X}}$ such that $M$ *completes* $E_t$, meaning that $M \cup E_t$ spans $\mathcal{X}$. Solving MFC for an initial forest where $\gamma(\mathcal{P}) = 1$ (e.g., obtained by terminating an exact algorithm early) produces an optimal MST (Figure 1b). Applying it to a initial forest with $\gamma(\mathcal{P}) > 1$ (Figure 1c) produces an approximately optimal spanning tree (Figure 1d).

MFC can be viewed as an MST problem defined over a complete *coarsened graph* $G_{\mathcal{P}} = (V_{\mathcal{P}}, E_{\mathcal{P}})$ where $V_{\mathcal{P}} = \{v_1, v_2, \ldots, v_t\}$ is the node set and $E_{\mathcal{P}} = \binom{V_{\mathcal{P}}}{2}$ is all pairs of nodes. Node $v_i$ corresponds to partition $P_i$ for each $i \in [t]$, and the weight between $v_i$ and $v_j$ is defined as the solution to the BCP problem between $P_i$ and $P_j$. Formally, the weight function $w^* : E_{\mathcal{P}} \to \mathbb{R}^+$ is given by

$$
w^*(v_i, v_j) = d(P_i, P_j).
\tag{2}
$$

Finding an MST of $G_{\mathcal{P}}$ with respect to $w^*$, and then mapping the edges in $G_{\mathcal{P}}$ back to the points in $\mathcal{X}$ that define the weight function $w^*$, solves the MFC problem. The challenge is that exactly computing $w^*$ can take $\Omega(n^2)$ distance queries, in particular when the component sizes are balanced.

**Existing MFC approximation.** Our previous MFC-Approx algorithm approximates MFC by considering only a subset of edges (Veldt et al., 2025). This algorithm selects one arbitrary representative point $r_i \in P_i$ for each $i \in [t]$, and completes the initial forest by adding only edges that are incident to one or two representatives. Conceptually this amounts to forming a new weight function $\hat{w} : V_{\mathcal{P}} \to \mathbb{R}^+$ such that $w^* \leq \hat{w}$, and then finding an MST in $G_{\mathcal{P}}$ with respect to $\hat{w}$. Previously we showed that this can be accomplished in $O(nt\mathcal{Q}_{\mathcal{X}})$ time (when $G_t$ is given), where $\mathcal{Q}_{\mathcal{X}}$ is the time to query one distance in $\mathcal{X}$. We proved that this algorithm returns a spanning tree that approximates the MFC problem to within a factor $(3 + \sqrt{5})/2 < 2.62$. Furthermore, it is a learning-augmented algorithm for the original metric MST problem with a parameter-dependent approximation guarantee of $(2\gamma + 1 + \sqrt{4\gamma + 1})/2 < (2\gamma + 1)$ (Veldt et al., 2025).

## 3   MULTI-REPRESENTATIVE MFC ALGORITHM

We present a generalization of MFC-Approx that selects a *set* of representatives for each component, rather than only one. For each $i \in [t]$, let $R_i \subseteq P_i$ be a nonempty subset of representatives for the $i$th component in $\mathcal{P}$. Let $R = \cup_{i=1}^{t} R_i$, and define $E_R = \{(r, x) : r \in R, x \in \mathcal{X}\}$. The new algorithm finds a minimum weight set of edges $\hat{M} \subseteq E_R$ to complete the initial forest. To do so, it finds an MST of the coarsened graph $G_{\mathcal{P}}$ with respect to a weight function $\hat{w} : E_{\mathcal{P}} \to \mathbb{R}^+$ given by

$$
\hat{w}(v_i, v_j) = \min \{d(P_i, R_j), d(P_j, R_i)\}.
\tag{3}
$$

For each pair $(v_i, v_j)$, the algorithm keeps track of the points $x, y \in \mathcal{X}$ such that $\hat{w}(v_i, v_j) = d(x, y)$, in order to map an MST in $G_{\mathcal{P}}$ back to the edge set $\hat{M} \subseteq E_R$. We denote this algorithm by MultiRepMFC($R$) or MultiRepMFC when $R$ is clear from context. By design, MultiRepMFC is a

simple way to interpolate between the existing MFC-Approx algorithm and an exact algorithm (when $R = \mathcal{X}$). Our key technical contributions are to provide an approximation analysis for this algorithm (Section 3.1), and present an approximately optimal strategy for selecting $R$ (Section 3.2).

## 3.1 Approximation analysis for fixed $R$.

To quantify the quality of spanning trees returned by MultiRepMFC($R$), define the *cost* of $P_i$ to be the maximum distance between any point in $P_i$ and its nearest representative:

$$\text{cost}(P_i, R_i) = \max_{x \in P_i} \min_{r \in R_i} d(x, r). \tag{4}$$

We extend this to a cost function on $\mathcal{P}$ by defining $\text{cost}(\mathcal{P}, R) = \sum_{i=1}^{t} \text{cost}(P_i, R_i)$. When $R$ is clear from context, we write $\text{cost}(P_i) = \text{cost}(P_i, R_i)$ and $\text{cost}(\mathcal{P}) = \text{cost}(\mathcal{P}, R)$. The following theorem uses this cost to define an instance-specific multiplicative approximation bound.

**Theorem 1.** *MultiRepMFC($R$) is an $\alpha$-approximation for MFC and an $(\alpha\gamma)$-approximation for metric MST where $\gamma$ is the overlap parameter for the initial forest and $\alpha = 1 + \text{cost}(\mathcal{P}, R)/w_{\mathcal{X}}(E_t)$.*

*Proof.* Let $T_{\mathcal{P}}^*$ denote an MST for the coarsened graph $G_{\mathcal{P}}$ with respect to $w^*$ as defined in Eq. (2). This $T_{\mathcal{P}}^*$ can be mapped to an edge set $M^* \subseteq \mathcal{X}$ that optimally solves MFC. Let $T^*$ be the spanning tree for $G_{\mathcal{X}}$ obtained by combining $M^*$ with the initial forest edges $E_t$. Thus,

$$w_{\mathcal{X}}(T^*) = w_{\mathcal{X}}(M^*) + w_{\mathcal{X}}(E_t) = w^*(T_{\mathcal{P}}^*) + w_{\mathcal{X}}(E_t). \tag{5}$$

Let $\hat{T}_{\mathcal{P}}$ be the MST in $G_{\mathcal{P}}$ with respect to $\hat{w}$ that MultiRepMFC finds, and $\hat{M}$ be the edge set in $\mathcal{X}$ it corresponds to. Then the spanning tree $\hat{T}$ returned by MultiRepMFC has weight

$$w_{\mathcal{X}}(\hat{T}) = w_{\mathcal{X}}(\hat{M}) + w_{\mathcal{X}}(E_t) = \hat{w}(\hat{T}_{\mathcal{P}}) + w_{\mathcal{X}}(E_t). \tag{6}$$

Since $T_{\mathcal{P}}^*$ is a tree, we can assign each edge in $T_{\mathcal{P}}^*$ to one of its endpoints in such a way that one node in $G_{\mathcal{P}}$ is assigned no edge, and every other node in $G_{\mathcal{P}}$ is assigned to exactly one edge of $T_{\mathcal{P}}^*$. This can be accomplished by selecting a node $v$ of degree 1 from $T_{\mathcal{P}}^*$, assigning $v$'s only incident edge to $v$, and then removing $v$ and its incident edge before recursing. This continues until there is only one node of $G_{\mathcal{P}}$ with no adjacent edges. We write $(v_i, v_j) \in T_{\mathcal{P}}^*$ to indicate an edge in this tree between $v_i$ and $v_j$ that is assigned to node $v_i$. Since each node is assigned at most one edge, we have that

$$\sum_{(v_i, v_j) \in T_{\mathcal{P}}^*} \text{cost}(P_i) \le \sum_{i=1}^{t} \text{cost}(P_i) = \text{cost}(\mathcal{P}). \tag{7}$$

For an arbitrary edge $(v_i, v_j) \in T_{\mathcal{P}}^*$, let $(x_a, x_b) \in P_i \times P_j$ be points in $\mathcal{X}$ defining the optimal edge weight $w^*(v_i, v_j) = d(x_a, x_b)$. Let $z \in R_i$ be the closest representative in $P_i$ to point $x_a$, meaning

$$d(x_a, z) = \min_{r \in R_i} d(x_a, r) \le \max_{x \in P_i} \min_{r \in R_i} d(x, r) = \text{cost}(P_i).$$

By definition, $\hat{w}(v_i, v_j)$ is at most the distance between $z$ and any point in $P_j$, which implies that $\hat{w}(v_i, v_j) \le d(z, x_b)$. Therefore

$$\hat{w}(v_i, v_j) \le d(z, x_b) \le d(x_a, x_b) + d(x_a, z) \le w^*(v_i, v_j) + \text{cost}(P_i). \tag{8}$$

Combining the bounds in (7) and (8) gives

$$\hat{w}(T_{\mathcal{P}}^*) = \sum_{(v_i, v_j) \in T_{\mathcal{P}}^*} \hat{w}(v_i, v_j) \le \sum_{(v_i, v_j) \in T_{\mathcal{P}}^*} [w^*(v_i, v_j) + \text{cost}(P_i)] \le w^*(T_{\mathcal{P}}^*) + \text{cost}(\mathcal{P}). \tag{9}$$

Putting these observations together proves the approximation for MFC:

$$
\begin{aligned}
w_{\mathcal{X}}(\hat{T}) = \hat{w}(\hat{T}_{\mathcal{P}}) + w_{\mathcal{X}}(E_t) && \text{(Eq. 6)} \\
\le \hat{w}(T_{\mathcal{P}}^*) + w_{\mathcal{X}}(E_t) && (\hat{T}_{\mathcal{P}} \text{ is optimal for } \hat{w}) \\
\le w^*(T_{\mathcal{P}}^*) + \text{cost}(\mathcal{P}) + w_{\mathcal{X}}(E_t) && \text{(Eq. 9)} \\
= w_{\mathcal{X}}(T^*) + \text{cost}(\mathcal{P}) && \text{(Eq. 5)} \\
\le \left(1 + \frac{\text{cost}(\mathcal{P})}{w_{\mathcal{X}}(E_t)}\right) w_{\mathcal{X}}(T^*) = \alpha w_{\mathcal{X}}(T^*)
\end{aligned}
$$

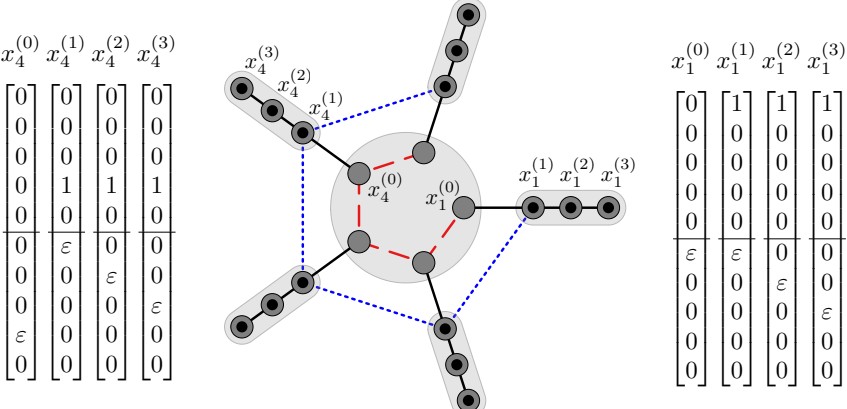

Figure 2: The MFC instance from Theorem 3 when $\ell = 3$ and $p = 5$. Initial forest edges are black solid lines. Points with black centers are representatives. Two points have distance $\varepsilon$ if they are in the same gray enclosing region, otherwise they have distance 1. Adding red dashed edges produces the optimal spanning tree. MultiRepMFC only adds edges incident to representatives, and therefore completes the forest with $\ell - 1$ edges of weight 1 (e.g., dotted blue edges).

where in the last step we have used the fact that $w_{\mathcal{X}}(E_t) \leq w_{\mathcal{X}}(T^*)$. To turn this bound into an $(\alpha\gamma)$-approximation for the original MST problem, it suffices to prove $w_{\mathcal{X}}(T^*) \leq \gamma w_{\mathcal{X}}(T_{\mathcal{X}})$, where $T_{\mathcal{X}}$ is an MST of $G_{\mathcal{X}}$ that leads to the smallest overlap parameter $\gamma$ for the initial forest. Let $I_{\mathcal{X}}$ denote the set of edges of $T_{\mathcal{X}}$ that are inside components $\mathcal{P}$, meaning that

$$w_{\mathcal{X}}(E_t) = \gamma w_{\mathcal{X}}(I_{\mathcal{X}}). \tag{10}$$

Furthermore, let $B_{\mathcal{X}} = T_{\mathcal{X}} \setminus I_{\mathcal{X}}$ be the set of edges in $T_{\mathcal{X}}$ that cross between components of $\mathcal{P}$. Observe that $B_{\mathcal{X}}$ must correspond to a spanning subgraph of the coarsened graph $G_{\mathcal{P}}$. If not, $T_{\mathcal{X}}$ would not provide a connected path between all pairs of components and hence would not span $G_{\mathcal{X}}$. The fact that $T_{\mathcal{P}}^*$ is a minimum weight spanner for $G_{\mathcal{P}}$ guarantees that $w^*(T_{\mathcal{P}}^*) \leq w_{\mathcal{X}}(B_{\mathcal{X}})$. Combined with (10), this gives the desired inequality:

$$w_{\mathcal{X}}(T^*) = w_{\mathcal{X}}(E_t) + w^*(T_{\mathcal{P}}^*) \leq \gamma w_{\mathcal{X}}(I_{\mathcal{X}}) + w_{\mathcal{X}}(B_{\mathcal{X}}) \leq \gamma w_{\mathcal{X}}(T_{\mathcal{X}}). \qquad \square$$

As a corollary, we improve on the previous analysis that proved MFC-Approx is a 2.62-approximation for MFC and a $(2\gamma + 1)$-approximation for metric MST.

**Corollary 2.** *MFC-Approx is a 2-approximation for MFC, and a $(2\gamma)$-approximation for MST where $\gamma$ is the overlap parameter for the initial forest.*

*Proof.* MFC-Approx is equivalent to MultiRepMFC when $R_i$ is a single arbitrary point from $P_i$ for each $i \in \{1, 2, \ldots, t\}$. We know $\text{cost}(P_i) \leq w_{\mathcal{X}}(T_i)$, since $\text{cost}(P_i)$ equals the distance between two specific points in $P_i$, and there is a path between these two points in $T_i$. Summing across all components gives $\text{cost}(\mathcal{P}) \leq w_{\mathcal{X}}(E_t)$. This in turn implies that $\alpha \leq 2$, proving the bound. $\qquad \square$

In addition to providing better approximation factors, our analysis is shorter and simpler than the prior analysis for MFC-Approx. See Appendix A for a more detailed comparison.

The following result shows that our approximation guarantees are tight in the worst-case. In particular, for a fixed number of $\ell$ representatives per component, the approximation factor in this theorem converges to $2 = 2\gamma$ as $p \to \infty$ and $\varepsilon \to 0$.

**Theorem 3.** *Let $p$ and $\ell$ be arbitrary positive integers, and $\varepsilon \in (0, 1)$ be arbitrary. There exists an initial forest with $\gamma(\mathcal{P}) = 1$ and a choice of $\ell$ representatives per component for which MultiRepMFC returns a tree that is a factor*

$$\frac{(2 + \ell\varepsilon - \varepsilon)p - 1}{(1 + \varepsilon\ell)p - \varepsilon}$$

*larger than the tree returned by optimally solving MFC (equivalent here to the metric MST problem).*

*Proof.* We will create an initial forest with $p$ components, each with $\ell + 1$ points, of which $\ell$ are representatives. All points will be represented by $(p + p')$-dimensional vectors, where $p' = \max\{\ell, p\}$. Let $x_i^{(j)}$ denote the $j$th point in the $i$th component, where $i \in [p]$ and $j \in \{0, 1, \ldots, \ell\}$. More precisely, for $i \in [p + p']$, let $\mathbf{e}_i$ be the $(p + p')$-dimensional vector that is all zeros except for a 1 in the $i$th entry. For $i \in [p]$ define $x_i^{(0)} = \varepsilon \cdot \mathbf{e}_{p+i}$ and $x_i^{(j)} = \mathbf{e}_i + \varepsilon \cdot \mathbf{e}_{p+j}$ for $j \in [\ell]$.

Let $R = \{x_i^{(j)} : i \in [t], j \in [\ell]\}$ denote the set of representatives. Let $S = \{x_i^{(0)} : i \in [t]\}$ be the remaining points, which we refer to as *small* points since they have norm $\varepsilon$. We consider the vector space $\mathcal{X} = R \cup S$ where distance is defined by the $\ell_\infty$ norm. This means that

$$d(x_i^{(j)}, x_a^{(b)}) = \|x_i^{(j)} - x_a^{(b)}\|_\infty = \begin{cases} \varepsilon & \text{if } i = a \text{ and } b > 0 \text{ and } j > 0 \\ \varepsilon & \text{if } b = j = 0 \\ 1 & \text{otherwise.} \end{cases}$$

In other words, the distance between two representatives in the same component is $\varepsilon$, the distance between two points in $S$ is $\varepsilon$, and the distance between every other pair of points is 1. See Figure 2.

To define the edges $E_t$ in the initial forest, for each $i \in [t]$ we add an edge from $x_i^{(j)}$ to $x_i^{(j+1)}$ for $j = 0, 1, \ldots, \ell - 1$. The first edge has weight 1, and the next $\ell - 1$ edges have weight $\varepsilon$. Thus, the weight of the initial forest is:

$$w(E_t) = p + p \cdot \varepsilon \cdot (\ell - 1).$$

Since $\varepsilon < 1$, the optimal solution to MFC is to add a spanning tree on small points. This costs $\varepsilon \cdot (p - 1)$, so if $T^*$ represents an optimal tree for the MFC problem,

$$w_\mathcal{X}(T^*) = p + p \cdot \varepsilon \cdot (\ell - 1) + \varepsilon(p - 1) = (1 + \varepsilon\ell)p - \varepsilon. \tag{11}$$

One can check that this tree is also optimal for the MST problem, so $\gamma(\mathcal{P}) = 1$.

MultiRepMFC only adds edges that are incident to one or more representatives, so this algorithm adds $p - 1$ edges of weight 1. Thus, if $\hat{T}$ is the tree returned by the algorithm, we have

$$w_\mathcal{X}(\hat{T}) = p + p \cdot \varepsilon \cdot (\ell - 1) + (p - 1) = (2 + \varepsilon\ell - \varepsilon)p - 1.$$

Taking the ratio $w_\mathcal{X}(\hat{T})/w_\mathcal{X}(T^*)$ yields the stated approximation factor. $\square$

Theorem 3 holds for a pathological construction and arbitrarily chosen representatives. By choosing representatives strategically, the approximation guarantee in Theorem 1 can be far better in practice.

## 3.2 THE BEST REPRESENTATIVES PROBLEM

We now focus on finding a set $R$ that optimizes the approximation ratio in Theorem 1. Let $b$ be a nonnegative *budget*, denoting the number of representatives $R$ is allowed to contain beyond having one representative per component. The Best Representatives problem (BESTREPS) is defined as:

$$\begin{aligned} \text{minimize} \quad & \text{cost}(\mathcal{P}, R) = \sum_{i=1}^t \max_{x \in P_i} \min_{r \in R_i} d(x, r) \\ \text{subject to} \quad & |R_i| \geq 1 \quad \forall i \in [t] \\ & \sum_{i=1}^t (|R_i| - 1) \leq b. \end{aligned} \tag{12}$$

If $t = 1$, this is equivalent to $k$-center with $k = b + 1$. Thus, BESTREPS is a generalization of $k$-center where there are multiple instances of points to cluster and the budget for cluster centers is shared across instances. Since the problem is NP-hard even for $t = 1$, it is impractical to solve optimally. However, we obtain a fast 2-approximation by combining an approximation algorithm for standard $k$-center (the $t = 1$ case) with a dynamic programming strategy for allocating budgets.[1]

**Greedy $k$-center for approximating allocation benefit.** For $i \in [t]$, we define

$$c_i^*(j) = \min_{R_i \,:\, |R_i| = j} \text{cost}(P_i, R_i) \quad \text{for } j \in [b + 1].$$

---

[1]An LLM was used to search for related work for this multi-instance $k$-center generalization, and also generated ideas for developing the 2-approximation algorithm for it. See Appendix B for details.

This captures the benefit for allocating $j$ representatives to cluster $P_i$. Computing $c_i^*(j)$ is equivalent to solving an NP-hard $k$-center problem on the set $P_i$ with $k = j$. We efficiently approximate this function for all $j \leq b+1$ by running the greedy 2-approximation of Gonzalez (1985) for $k$-center with $k = b+1$. This method starts by choosing an arbitrary first cluster center. At iteration $j \leq k$, it chooses the $j$th cluster center to be the point that is farthest away from the first $j-1$ cluster centers. Let $R_{i,j}$ be the the first $j$ cluster centers found by this procedure, and define

$$\hat{c}_i(j) = \text{cost}(P_i, R_{i,j}) \quad \text{for } j \in [b+1].$$

By the algorithm's 2-approximation guarantee, we know $\hat{c}_i(j) \leq 2c_i^*(j)$ for $i \in [t]$ and $j \in [b+1]$.

**DP for allocating representatives.** We allocate representatives to components by solving

$$\text{minimize} \quad \sum_{i=1}^{t} \hat{c}_i(b_i + 1) \quad \text{subject to} \sum_{i=1}^{t} b_i = b \text{ and } b_i \geq 0 \quad \forall i \in [t], \tag{13}$$

where $b_i \geq 0$ represents the number of *extra* representatives assigned to $P_i$. This is a variant of the knapsack problem where the objective function is nonlinear, all items have weight 1, and we allow repeat items ($b_i \geq 1$). If the $\hat{c}_i$ functions are already computed, this can be solved optimally in $O(tb^2)$ time via dynamic programming (DP). The DP approach for problems in this form is standard. We provide full details in Appendix B for completeness, as well as a proof for the following result.

**Theorem 4.** *Let $\{\hat{b}_i \colon i \in [t]\}$ be the optimal solution to Problem* (13). *For $i \in [t]$, define $R_i$ to be the first $\hat{b}_i + 1$ cluster centers chosen by running the greedy 2-approximation for $k$-center on $P_i$. Then $\{R_i \colon i \in [t]\}$ is a 2-approximate solution for* BESTREPS.

### 3.3 ALGORITHM VARIANTS AND RUNTIME ANALYSIS

We now summarize several different approximation algorithms for MFC (and their runtimes) that are obtained by combining MultiRepMFC with different strategies for finding $R$. See Appendix C for more details. We assume $t = O(n^\delta)$ for some $\delta \in [0, 1)$. Let DP-MultiRepMFC denote the algorithm that runs MultiRepMFC after finding a set of representatives using the dynamic programming strategy from Theorem 4. It has a runtime of $O(n\mathcal{Q}_\mathcal{X}(b+t) + tb^2)$. Greedy-MultiRepMFC is a faster approach that greedily allocates representatives to components iteratively in a way that leads to the best improvement to the objective in Problem 13 at each step. It has a runtime of $O(n\mathcal{Q}_\mathcal{X}(b+t))$. Fixed($\ell$)-MultiRepMFC is a simple baseline that chooses $\ell \geq 1$ representatives per component by running the greedy 2-approximation for $k$-center on each component with $k = \ell$. It has a runtime of $O(n\mathcal{Q}_\mathcal{X}(b+t))$, but only applies to budgets $b$ that are multiples of $t$. All three of these algorithms are 2-approximations for MFC (and learning-augmented $2\gamma$-approximations for metric MST) by Theorem 1. The asymptotic bottleneck for all three runtimes is computing $\hat{w}$. Fixed($\ell$)-MultiRepMFC and Greedy-MultiRepMFC are faster than DP-MultiRepMFC, but DP-MultiRepMFC is the only method that satisfies an approximation guarantee for the BESTREPS step. The runtimes above assume that the initial forest is already given. If one factors in the time it takes to compute the initial forest, choosing representatives constitutes an even smaller portion of the runtime.

## 4 EXPERIMENTS

Prior work has already shown that the MFC framework (which includes both computing an initial forest and running MFC-Approx) is fast and finds nearly optimal spanning trees for a wide range of dataset types and metrics. Since our work focuses on improved algorithms for the MFC step, our experiments also focus on this step, rather than on again comparing the entire MFC framework against an exact $\Omega(n^2)$-time algorithm for metric MST. We specifically address the following questions relating to MultiRepMFC, our approximation bound $\alpha$, and strategies for choosing representatives.

*Question 1:* How does MultiRepMFC compare (in terms of runtime and spanning tree cost) against the MFC-Approx algorithm ($b = 0$) and the $\Omega(n^2)$ algorithm for the MFC step ($b = n$)?

*Question 2:* What is the runtime vs. quality tradeoff between using different strategies for approximating the BESTREPS step in practice (Dynamic, Greedy, Fixed($\ell$))?

*Question 3:* How does our instance-specific approximation bound ($\alpha$ in Theorem 1) compare to the worst-case 2-approximation and the actual approximation achieved in practice?

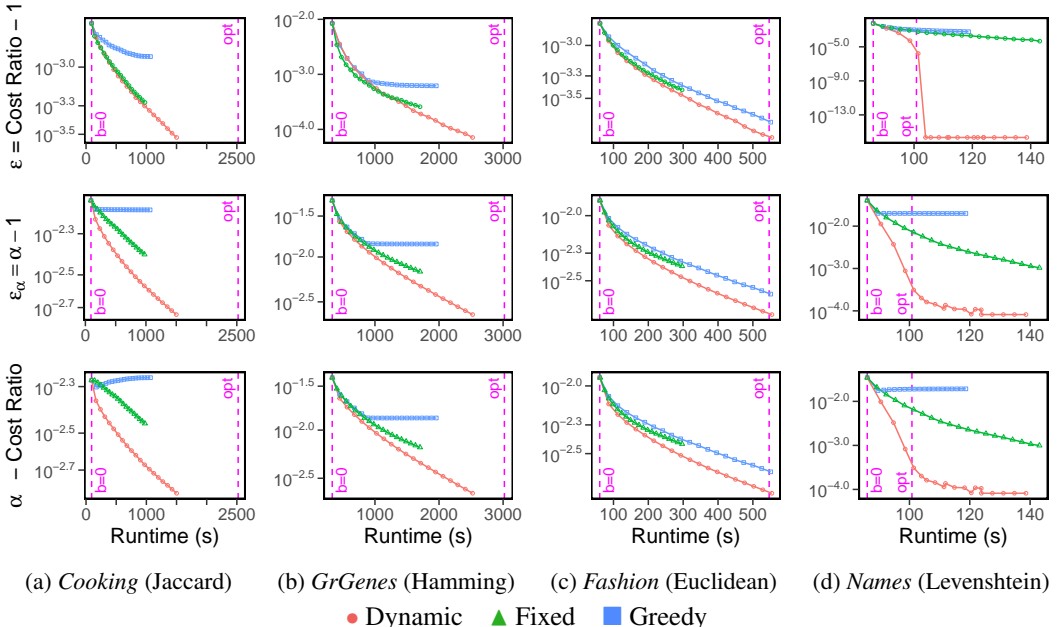

(a) *Cooking* (Jaccard)   (b) *GrGenes* (Hamming)   (c) *Fashion* (Euclidean)   (d) *Names* (Levenshtein)

● Dynamic   ▲ Fixed   ■ Greedy

Figure 3: We display the performance of each variant of MultiRepMFC as runtime increases. Each point corresponds to running one method with a fixed budget $b$. The top row shows the value of $\varepsilon$ such that a method obtains a $(1 + \varepsilon)$-approximation in practice. The second row shows the value $\varepsilon_\alpha$ such that we can guarantee a $(1 + \varepsilon_\alpha)$-approximation using Theorem 1. Computing $\varepsilon_\alpha$ is fast. Computing $\varepsilon$ is impractical as it requires optimally solving MFC. The last row shows the gap between $\alpha$ and the true approximation as runtime increases. We see that all variants of MultiRepMFC provide a useful interpolation between the existing MFC-Approx algorithm ($b = 0$ vertical dashed line) and an optimal MFC algorithm (right vertical dashed line). All plots also show that dynamic programming produces better true approximations (top row), much better approximation bounds (middle row), and is faster at shrinking the gap between the bound and true approximation (last row). For *Cooking*, 16 random orderings of the entire dataset ($n = 39,774$) were used, for all others we take 16 uniform random samples of size $n = 30,000$. Average results are then displayed.

**Implementation details and experimental setup.** Our algorithm implementations are in C++ and directly build on the open-source code made available for MFC-Approx in our prior work (Veldt et al., 2025). We also apply a similar experimental setup. We compute initial forests by partitioning $\mathcal{X}$ using a $k$-center algorithm and then finding optimal MSTs for partitions. We choose $t = \sqrt{n}$ partitions since this approximately minimizes the asymptotic runtime for computing a spanning tree (when including the time to form the initial forest); see Appendix C for more details. We consider 4 datasets also used previously (Veldt et al., 2025), chosen since each corresponds to a different dataset type and distance metric. These are: *Cooking* (set data; Jaccard distance), *GreenGenes* (fixed-length sequences; Hamming distance), *FashionMNIST* (784-dimensional points, Euclidean distance), and *Names-US* (strings; Levenshtein edit distance). See Appendix D for more details on datasets.

To address our three questions, we run DP-MultiRepMFC, Greedy-MultiRepMFC, and Fixed($\ell$)-MultiRepMFC for a range of budgets $b$. For our comparisons, we also run MFC-OPT: an optimal algorithm for MFC that finds an MST of the coarsened graph with respect to the optimal weight function $w^*$. Each run of each algorithm produces a spanning tree that completes the initial forest. To measure spanning tree quality, we compute the *Cost Ratio* for MFC: the weight of the spanning tree produced by the algorithm divided by the weight of the tree produced by MFC-OPT. We also compute $\alpha$ from Theorem 1, which is an upper bound for *Cost Ratio*. This bound $\alpha$ differs for each algorithm and choice of $b$, since it depends on how well the BESTREPS step is solved.

The first row of Figure 3 displays *Cost Ratio* $- 1$ versus runtime for each algorithm. Note that if $\varepsilon = $ *Cost Ratio* $- 1$, this means the algorithm achieved a $(1 + \varepsilon)$-approximation. For the $x$-axis, the runtime includes the time for approximating BESTREPS plus the time for MultiRepMFC. The second row of plots displays $\varepsilon_\alpha = \alpha - 1$ in the $y$-axis. This shows us the value of $\varepsilon_\alpha$ for which we can

*guarantee* an algorithm has achieved at least a $(1 + \varepsilon_\alpha)$-approximation, by Theorem 1. The third row of plots displays $\alpha - Cost\ Ratio$, which is the gap between our bound on the approximation factor and the true approximation factor. In Appendix D, we show results for all these metrics as the budget $b$ varies. However, this does not provide as direct of a comparison, since the runtime for each method depends differently on $b$. Here in the main text, we primarily focus on understanding how well each method performs within a fixed runtime budget (rather than fixed $b$).

**Comparing against MFC-Approx and MFC-OPT (Question 1).** When $b = 0$ (leftmost point in each plot), all MultiRepMFC algorithms correspond to the previous MFC-Approx algorithm. The output for each algorithm traces out a performance curve as $b$ (and runtime) increases. These curves tend to decrease steeply at the beginning, showing that MultiRepMFC produces noticeably better spanning trees than MFC-Approx with only a small amount of extra work. In many cases, the spanning tree quality gets very close to an optimal solution at a fraction of the time it takes to run MFC-OPT. One outlier in these results is the Names-US dataset, where MFC-OPT is much faster than usual. This is because initial forests for Names-US are highly imbalanced, with one large component containing nearly all the points. For highly-imbalanced forests, it is much cheaper to optimally solve the MFC step. Running MultiRepMFC is therefore not useful for large values of $b$. Nevertheless, for small values of $b$, MultiRepMFC provides a meaningful interpolation between MFC-Approx and MFC-OPT.

**Comparing methods for BESTREPS (Question 2).** From the top row of Figure 3, we see that DP-MultiRepMFC tends to produce the best spanning trees within a fixed time budget. Perhaps surprisingly, the simplest method Fixed($\ell$)-MultiRepMFC tends to outperform Greedy-MultiRepMFC, whose progress tends to plateau after a certain point. This may be because Greedy-MultiRepMFC is too myopic in assigning representatives. For example, it is possible that adding one extra representative to a certain component would change the objective very little, but adding two or more would significantly decrease the objective. Fixed($\ell$)-MultiRepMFC would be able to achieve this benefit for the right choice of $\ell$, whereas Greedy-MultiRepMFC may never notice the benefit.

**Comparing $\alpha$ values (Question 3).** Our bound $\alpha$ (second row of plots in Figure 3) is always very close to 1, and provides a much better bound to the true approximation ratio than the worst-case 2-approximation. This can be seen in the third row of plots in Figure 3. This is significant since computing the true *Cost Ratio* is impractical, as it requires optimally solving MFC. However, $\alpha$ can be computed easily in the process of running MultiRepMFC, and therefore serves as a very good proxy for the true approximation ratio with virtually no extra effort. As a practical benefit, this opens up the possibility of choosing $b$ dynamically in practice. In particular, one can choose to add representatives until achieving a satisfactory value of $\alpha$, and only then run MultiRepMFC.

Figure 3 also shows that different approaches for BESTREPS perform differently in terms of how well they minimize $\alpha$. While DP-MultiRepMFC is slightly better than other methods in terms of *Cost Ratio*, it is far better at minimizing $\alpha$. Again, this is significant because $\alpha$ is an approximation guarantee that we can efficiently obtain in practice, unlike the true *Cost Ratio*. Furthermore, as runtime increases, the gap between $\alpha$ and *Cost Ratio* shrinks more quickly for DP-MultiRepMFC than for other methods (Figure 3, third row). This provides further evidence for the benefits of the dynamic programming approach for BESTREPS, which helps further address Question 2.

## 5 CONCLUSIONS AND DISCUSSION

Metric forest completion is a learning-augmented framework for finding a spanning tree in an arbitrary metric space, when the learned input is an initial forest that serves as a starting point. We have introduced a generalized approximation algorithm for this problem that comes with better theoretical approximation guarantees, which we prove are tight. Our results also include very good instance-specific approximation guarantees that overcome worst-case bounds. In numerical experiments, we show that with a small amount of extra work, we can obtain much better quality solutions for MFC than the prior approach. One open direction is to pursue approximations for metric MST in terms of other quality parameters (aside from $\gamma$-overlap) for the initial forest. Another question is whether one can achieve worst-case approximation factors below 2 for MFC, using alternative techniques with subquadratic complexity. Finally, an interesting question is whether we can prove general lower bounds on the approximation ratio that hold for all algorithms with subquadratic complexity.

ACKNOWLEDGEMENTS

This work was performed under the auspices of the U.S. Department of Energy by Lawrence Livermore National Laboratory under Contract DE-AC52-07NA27344 (LLNL-CONF-2011636), and was supported by LLNL LDRD project 24-ERD-024.

REPRODUCIBILITY STATEMENT

Our code is included in the paper supplement and also publicly available at `https://github.com/tommy1019/MetricForestCompletion`. This includes all source code for our algorithms, the commands that were run to produce the main results, and scripts for plotting our results. The output from our experiments is included in a results folder, so that all plots from the main text can be reproduced. Most of the datasets are too large to include in the supplementary file, so we have included instructions in the supplement's README regarding where the original datasets can be obtained and how they were preprocessed. The appendix of our paper also includes a description for each dataset and references to the original sources. For our theoretical results, complete proof details are included either in the main text or the appendix.

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

---

**Algorithm 1** MultiRepMFC($R = \{R_i \colon i \in [t]\}$)

---

1: **Input:** $\mathcal{X} = \{x_1, x_2, \ldots, x_n\}$, components $\mathcal{P} = \{P_1, P_2, \ldots, P_t\}$, spanning trees $\{T_1, T_2, \ldots, T_t\}$, nonempty $R_i \subseteq P_i$ for each $i \in [t]$.
2: **Output:** Spanning tree for $G_{\mathcal{X}} = (\mathcal{X}, E_{\mathcal{X}})$.
3: **for** $(i, j) \in \binom{t}{2}$ **do**
4:    $w_{i \to j} = \min_{x_i \in P_i, r_j \in R_j} d(x_i, r_j)$
5:    $w_{j \to i} = \min_{x_j \in P_j, r_i \in R_i} d(x_j, r_i)$
6:    $\hat{w}_{ij} = \min\{w_{i \to j}, w_{j \to i}\}$
7: **end for**
8: $\hat{T}_{\mathcal{P}} = \mathsf{OptMST}(\{\hat{w}_{ij}\}_{i,j \in [t]})$
9: Return spanning tree $\hat{T}$ obtained by combining $E_t$ with edges corresponding to $\hat{T}_{\mathcal{P}}$.

---

## A   ADDITIONAL MULTIREPMFC ALGORITHM DETAILS

Pseudocode for MultiRepMFC is given in Algorithm 1. This relies on a black-box function OptMST which computes an optimal solution for a minimum spanning tree for a graph. This is specifically applied to find an MST of the coarsened graph with respect to the new weight function $\hat{w} \colon V_{\mathcal{P}} \to \mathbb{R}$, which corresponds to a set of edges in $\mathcal{X}$ that completes the initial forest $G_t = (\mathcal{X}, E_t)$. When $|R_i| = 1$ for every $i \in [t]$, this corresponds to our previous MFC-Approx algorithm (Veldt et al., 2025).

**Comparison between Theorem 1 and prior work.** Our proof that MFC-Approx is a 2-approximation for MFC (and a $2\gamma$-approximation for metric MST) simplifies our prior analysis (Veldt et al., 2025) in a few ways. The prior analysis relied on partitioning edges in the coarsened graph based on whether or not they were $\beta$-bounded (meaning $\hat{w}_{ij} \leq \beta w_{ij}^*$) for some initially unspecified $\beta > 1$. The analysis then considered two other spanning trees of the coarsened graph that are optimal with respect to two other hypothetical weight functions that depend on $\beta$. This led to bounds for different parts of the weight of $\hat{T}$ (the tree returned by MFC-Approx), in terms of different expressions involving $\beta$. The best approximation guarantee of $\beta = (3 + \sqrt{5})/2$ was obtained by solving a quadratic equation resulting from the bounds.

In contrast, our new analysis completely avoids the need to partition edges based on an unknown $\beta$, work with hypothetical weight functions, or solve for the best $\beta$ in this way. Our analysis shares some other steps in common with our prior proof (Veldt et al., 2025), but is ultimately able to apply basic facts about distances and triangle inequalities more directly. This leads to an analysis that is simpler, shorter, and tighter.

## B   DYNAMIC PROGRAMMING FOR REPRESENTATIVE ALLOCATION

Consider a resource allocation problem of the form

$$\text{minimize} \quad \sum_{i=1}^{t} f_i(\mathbf{b}[i]) \tag{14}$$

$$\text{subject to} \quad \sum_{i=1}^{t} \mathbf{b}[i] = b \tag{15}$$

$$\mathbf{b} \in \mathbb{N}^t \tag{16}$$

where we assume the functions $\{f_i \colon i \in [t]\}$ are given and we use the convention that natural numbers include zeros: $\mathbb{N} = \{0, 1, 2, \ldots\}$. This matches Problem (13) in the main text after applying a change of function $f_i(j) = \hat{c}_i(j + 1)$ to better highlight that the goal is to assign *extra* representatives, beyond the first representative for each component. The following theorem guarantees that if we can solve this problem, we can use it to obtain a 2-approximation for BESTREPS.

**Theorem 4.** Let $\{\hat{b}_i \colon i \in [t]\}$ be the optimal solution to Problem (13). For $i \in [t]$, define $R_i$ to be the first $\hat{b}_i + 1$ cluster centers chosen by running the greedy 2-approximation for $k$-center on $P_i$. Then $\{R_i \colon i \in [t]\}$ is a 2-approximate solution for BESTREPS.

*Proof.* BESTREPS is equivalent to minimizing $\sum_{i=1}^{t} c_i^*(b_i + 1)$ subject to $\sum_{i=1}^{t} b_i = b$ and $b_i \geq 0$ for every $i \in [t]$. Let $\{b_i^* : i \in [t]\}$ denote an optimal solution for this problem. The 2-approximation guarantee for $\{\hat{b}_i : i \in [t]\}$—and the corresponding sets $\{R_i\}$—follows from the fact that $\{\hat{b}_i\}$ are optimal for the $\{\hat{c}_i\}$ functions, and the fact that $c_i^*(j) \leq \hat{c}_i(j) \leq 2c_i^*(j)$ for every $i \in [t]$ and $j \in [b+1]$:

$$\sum_{i=1}^{t} c_i^*(\hat{b}_i + 1) \leq \sum_{i=1}^{t} \hat{c}_i(\hat{b}_i + 1) \leq \sum_{i=1}^{t} \hat{c}_i(b_i^* + 1) \leq 2\sum_{i=1}^{t} c_i^*(b_i^* + 1).$$

$\square$

We can solve the problem in (14) using dynamic programming. For integers $B \in [0, b]$ and $T \in [t]$, define $\Omega_T^B = \{\mathbf{b} \in \mathbb{N}^T : \sum_{i=1}^{T} \mathbf{b}[i] = B\}$ and define

$$F(T, B) = \min_{\mathbf{b} \in \Omega_T^B} \sum_{i=1}^{T} f_i(\mathbf{b}[i]).$$

Our goal then is to efficiently compute $F(t, b)$.

In the context of the BESTREPS problem, $F(T, B)$ is the optimal way to assign $B$ extra representatives to the first $T$ components. If there are no extra representatives to assign, we can see that

$$F(T, 0) = \sum_{i=1}^{T} f_i(0) \quad \text{for } T \in [t].$$

If there is only one component to assign extra representatives to, then we have

$$F(1, B) = f_1(B) \quad \text{for } B = 0, 1, \ldots, b.$$

Observe next that the optimal way to allocate $B$ representatives across the first $T$ components is found by considering all ways to optimally allocate $k$ representatives to the first $T - 1$ components, while allocating $B - k$ representatives to the $T$th component. This is captured by the formula:

$$F(T, B) = \min_{0 \leq k \leq B} F(T - 1, k) + f_T(B - k).$$

Using a bottom-up dynamic programming algorithm, computing $F(T, B)$ when given $F(T - 1, k)$ and $f_T(B - k)$ for every $k \in [0, B]$ takes $O(B) = O(b)$ time for every $T \leq t$. Since we need to compute $F(T, B)$ for $t$ choices of $T$ and $b$ choices of $B$, the overall runtime is $O(tb^2)$.

In practice, we need to know not just the value of $F(t, b)$ but the choice of $\mathbf{b} \in \mathbb{N}^t$ that produces the optimal solution, since this determines the number of representatives for each component. We can accomplish this naively by storing a vector of length-$t$ for each choice of $F(T, B)$, leading to a memory requirement of $O(t^2 b)$. In practice, we reduce this to $O(tb)$ by noting that $F(T, B)$ only depends on $F(T - 1, k)$ for $k \in [0, B]$. Thus, as long as we save the length-$t$ vectors associated with $F(T - 1, k)$ for $k \in [0, B]$, we can discard all length-$t$ vectors associated with $F(J, k)$ for $J < T - 1$ and $k \in [0, B]$.

**Details on the use of LLMs for the BESTREPS results.** Given the similarity between BESTREPS and the classical $k$-center problem, we prompted an LLM to help check whether this problem had been previously studied. The LLM noted several other variants of $k$-center and similar resource allocation problems, but was unable to find prior examples where this exact problem had been studied. The LLM then suggested a greedy algorithm that it claimed was a 2-approximation algorithm for this problem, but the approximation analysis it provided was incorrect. With additional prompting, the LLM suggested a dynamic programming approach. Although the LLM's dynamic programming algorithm and its proof still contained minor errors, the general strategy matched the basic approach we ultimately used to prove a 2-approximation. This is a natural strategy for an LLM to suggest, given that the dynamic programming approach is standard for variants of the knapsack problem and resource allocation problems in this form. See, for example, the work of Marsten & Morin (1978), which effectively covers the same strategy. LLMs were not used in research ideation for any other aspects of the paper, and in particular were not used for any of the design or analysis in Section 3.1. LLMs were also not used to aid in the final write-up of results for BESTREPS, or for the write-up of any other section of the paper.

## C  DETAILS FOR ALGORITHM VARIANTS AND RUNTIMES

There are several nuances to consider when reporting runtimes for our algorithm variants DP-MultiRepMFC, Greedy-MultiRepMFC, and Fixed($\ell$)-MultiRepMFC. These all have different runtimes when addressing the BESTREPS problem, but in many cases those runtimes are overshadowed by the MultiRepMFC step that follows when approximating MFC. The relative difference between these algorithms can be further obscured if one also considers the time it takes to compute the initial forest. Although computing an initial forest is not part of the MFC problem, it is an important consideration if the ultimate goal is to approximate the original metric MST problem.

We provide a careful runtime comparison for each algorithm here, along with some considerations regarding the time to compute the initial forest. Let $\mathbb{Q}_{\mathcal{X}}$ denote the time for one distance query in $\mathcal{X}$. We assume the number of components in the initial forest is $t = O(n^\delta)$ for some $\delta \in [0, 1)$, since the MFC framework only leads to subquadratic time algorithms if $t$ is dominated by the number of points. We use $\tilde{O}$ to hide logarithmic factors in $n$.

**Runtimes for BESTREPS step.** In order to find a set of representatives $R$ of size $b + t$ using the dynamic programming algorithm or the greedy method, we first must compute $\hat{c}_i(j)$ for $i \in [t]$ and $j \in [b+1]$. To do so, we run the standard greedy 2-approximation for $k$-center on $P_i$ for each $i \in [t]$ with $k = b + 1$. This means $k|P_i|$ distance queries for $i \in [t]$, so this step takes $O((b+1)n\mathbb{Q}_{\mathcal{X}})$ time after summing over all $i \in [t]$.

*Dynamic programming.* The dynamic programming step takes an additional $O(tb^2)$ time to allocate representatives to components. The total time for approximating BESTREPS via dynamic programming is therefore $O(n(b+1)\mathbb{Q}_{\mathcal{X}} + tb^2)$.

*Greedy representative allocation.* The greedy algorithm for BESTREPS starts with $(b_1, b_2, \ldots, b_t) = (0, 0, \ldots, 0)$. It then simply iterates through the number of additional representatives from 1 to $b$, and at each step adds 1 to the $b_i$ value that leads to the largest decrease in the objective $\sum_{i=1}^{t} \hat{c}_i(b_i + 1)$. More concretely, the algorithm must maintain the value of

$$\Delta_i = \hat{c}_i(b_i + 1) - \hat{c}_i(b_i + 2)$$

for each $i \in [t]$, and choose the component with maximum $\Delta_i$ at each step. Observe that $\Delta_i \geq 0$ for each $i \in [t]$, since $\hat{c}_i$ is a decreasing cost function. A simple $O(tb)$-time implementation is to store $\Delta_i$ values in an array and iterate through all $t$ values to find the maximum at each step. This can be improved to $O(t + b \log t)$ time using a heap. However, in either case this step is dominated by the time to compute the $\hat{c}_i$ functions. So the runtime for the greedy algorithm for BESTREPS is $O(n(b+1)\mathbb{Q}_{\mathcal{X}})$.

*Fixed($\ell$)-MultiRepMFC baseline.* In order to choose $\ell = b/t$ representatives per component (assuming $b$ is a multiple of $t$), we simply run the greedy $k$-center 2-approximation on each component with $k = \ell$. We then use the resulting centers as the representatives. This has a faster runtime for BESTREPS of $O(\ell n \mathbb{Q}_{\mathcal{X}})$.

**Runtime for MultiRepMFC step.** Fix a set $\{R_i : i \in [t]\}$ of nonempty representative sets for the components. Let $R = \bigcup_{i=1}^{t} R_i$ where $|R| = b + t$. Given this input, MultiRepMFC first computes the distance between each $r \in R_i$ and every $x \in \mathcal{X} \setminus P_i$. This is a total of

$$\sum_{i=1}^{t} |R_i| \cdot (|\mathcal{X}| - |P_i|) < (b + t)n$$

distance queries, needed to define the weight function $\hat{w}$ for the coarsened graph. It then takes $O(t^2 \log t)$ time to find the MST of the coarsened graph with respect to $\hat{w}$ using Kruskal's algorithm. One could implement the latter step more quickly using alternate MST techniques, but our implementations apply Kruskal's since this is simple and is not the bottleneck of MultiRepMFC, neither in theory nor practice. Overall, running MultiRepMFC with a fixed $R$ takes $O(n(b+t)\mathbb{Q}_{\mathcal{X}})$.

**Full runtime analysis for MFC approximation algorithms.** Although Greedy-MultiRepMFC and Fixed($\ell$)-MultiRepMFC differ in terms of their runtime for the BESTREPS step, both of these algorithms are asymptotically dominated by MultiRepMFC. Therefore, as approximation algorithms for MFC (i.e., ignoring initial forest computation time) their runtime is $O(n(b + t)\mathbb{Q}_{\mathcal{X}})$. DP-MultiRepMFC, on the other hand, has a runtime of $O(n(b + t)\mathbb{Q}_{\mathcal{X}} + tb^2)$ to account for the more

accurate (but slower) dynamic programming strategy for allocating representatives. In cases where $b$ is small enough ($b = O(n/t)$), this term is negligible asymptotically. Hence, when we have a small budget for additional representatives, we would expect DP-MultiRepMFC to improve over MFC-Approx and Greedy-MultiRepMFC (note that Fixed($\ell$)-MultiRepMFC is not defined in this case). As $b$ increases beyond this limit, we still expect DP-MultiRepMFC to produce better spanning trees than its competitors when we consider a fixed $b$, but the runtimes are then not directly comparable.

**Considerations for computing the initial forest.** The runtime for computing an initial forest can vary significantly depending on the strategy used. In our prior work (Veldt et al., 2025) we computed initial forests by first running the greedy 2-approximation for $k$-center with $k = t$ to partition $\mathcal{X}$, and then applying Kruskal's algorithm to find optimal MSTs of the components. In cases where the $k$-center step produces balanced clusters (which it often did in practice but is not guaranteed to), this runs in $\tilde{O}(nt\mathcal{Q}_{\mathcal{X}} + n^2/t)$-time. In practice, we found that this was typically much faster than running an $\Omega(n^2)$-time exact algorithm for MST, but was also usually the bottleneck for the MFC framework. In particular, for most values of $t$, computing an initial forest was slower than running MFC-Approx. Whether or not computing the initial forest is the most expensive step for finding an approximate spanning tree, this will have an impact on the comparison between DP-MultiRepMFC, Greedy-MultiRepMFC, and Fixed($\ell$)-MultiRepMFC. Especially in cases where computing an initial forest step is expensive, the runtime differences between these algorithms for the MFC will be less important.

**Motivation for $t = \sqrt{n}$.** In our numerical experiments, we set $t = \sqrt{n}$ since this roughly minimizes the asymptotic runtime when factoring in the initial forest computation. In more detail, consider simplified conditions where the initial $k$-center step produces balanced partitions, $b = O(t)$, and $\mathcal{Q}_{\mathcal{X}} = O(\log n)$. If these conditions hold, the runtime for finding an initial forest and running DP-MultiRepMFC is $\tilde{O}(nt + t^3 + n^2/t)$. This is minimized when $t = \Theta(\sqrt{n})$. If we instead used Greedy-MultiRepMFC or Fixed($\ell$)-MultiRepMFC, the runtime would be $\tilde{O}(nt + n^2/t)$ under these conditions, which is still minimized by $t = \Theta(\sqrt{n})$. Even if these conditions do not all perfectly hold, we expect $t = \Theta(\sqrt{n})$ to at least approximately minimize the runtime. We remark finally that there are other existing heuristics for computing an approximate MST that also rely on partitioning a dataset into $t$ components and then connecting disjoint components. Using a similar arguments, these methods select $t = \sqrt{n}$ in order to minimize the overall runtime (Jothi et al., 2018; Zhong et al., 2015).

**Comparison with existing EMST algorithms.** Although our main focus is to design spanning tree algorithms that work for arbitrary metric spaces, it is also useful to consider the guarantees of MultiRepMFC against existing techniques that work for the more restrictive Euclidean minimum spanning tree problem (EMST). For this comparison, we consider $d$-dimensional vectors where $d$ is a constant but potentially very large. Using the simplified analysis above, the runtime of MultiRepMFC scales roughly as $n^{1.5}$ by choosing $t = \Theta(\sqrt{n})$.

As noted above, there are several existing heuristics for EMST that partition the data into $\sqrt{n}$ components and then add additional edges to connect components before refining into an overall spanning tree (Jothi et al., 2018; Zhong et al., 2015). MultiRepMFC has a comparable asymptotic runtime to these methods, while also having the advantage of satisfying concrete theoretical guarantees.

Agarwal et al. (1990) showed that an optimal EMST can be computed in $O(n^{2 - \frac{2}{\lceil d/2 \rceil + 1} + \varepsilon})$ time. A simple calculation shows that this is slower than $O(n^{1.5})$ whenever $d \geq 6$. Thus, although it provides optimal solutions, it is not expected to be nearly as scalable as MultiRepMFC for even for modest values of $d$. Arya & Mount (2016) presented a method for computing a $(1 + \varepsilon)$-approximate EMST in time $O(n \log n + \varepsilon^{-2} \log^2 \frac{1}{\varepsilon})$ if $d$ is a constant. However, this runtime hides exponential factors in $d$ of the form $O(1)^d$, which likely be a challenge for practical applications in situations where $d$ is large, e.g., the FashionMNIST dataset where $d = 784$. In contrast, MultiRepMFC has no exponential dependence on $d$, and has no problems being run on FashionMNIST and other high-dimensional Euclidean datasets.

There are many other theoretical and practical algorithms for EMSTs that have been designed under diverse computational models and assumptions (Chen et al., 2023; 2022; March et al., 2010; Wang et al., 2021; Jayaram et al., 2024). The size of the literature complicates a comprehensive comparison. We expect that the best practical algorithms for EMSTs will outperform MultiRepMFC on

Euclidean data, since they are specialized to this setting and heavily leverage the assumption that the data is Euclidean. Nevertheless, MultiRepMFC also provides a scalable approach that is also very simple to implement. Its most important feature is that it applies to arbitrary metrics.

## D    ADDITIONAL EXPERIMENTAL DETAILS

Our experiments are run on a large research server with 1TB of RAM with two 32-Core AMD processors. We use a subset of the datasets considered in ou prior work (Veldt et al., 2025). Specifically, we selected four datasets that correspond to different distance metrics. We summarize the datasets below, along with a link to the original data source(s). See our prior work (Veldt et al., 2025) for additional steps in preprocessing data from the original source.

- *Cooking (Jaccard distance)* (Kaggle, 2015; Amburg et al., 2020). Sets of food ingredients that define recipes. There are $n = 39,774$ recipes and 6714 ingredients.
- *FashinMNIST (Euclidean distance)* (Xiao et al., 2017). Vectors of size $d = 784$ representing flattened images of size $28 \times 28$ pixels, where each image is a picture of a clothing item.
- *Names-US (Levenshtein edit distance)* (Remy, 2021). Each data point is a string representing a last name of someone in the United States. The average name length is 6.67.
- *GreenGenes-aligned (Hamming distance)* (DeSantis et al., 2006). An alternate form for the GreenGenes dataset where sequences have been aligned so that each is represented by a fixed length sequence of 7682 characters.

When performing experiments for the *Cooking* dataset, we repeatedly take uniform random orderings of data points. Because our algorithms are deterministic, this random ordering effects only the arbitrary elements selected for the first center during the $k$-center step of the initial forest computation, and the first representative chosen for each component when approximating the BESTREPS problem. For all other datasets, we take uniform random samples of $n = 30,000$ data points. In all cases, we choose $t = \lfloor \sqrt{n} \rfloor$, which equals 199 for *Cooking* and 173 for other datasets. In all our plots, we report the average performance over 16 different random samples. Although our approximation algorithms can scale to much larger sizes of $n$, we also run an exact algorithm for MFC for our comparisons, which can take $\Omega(n^2)$ in the worse case. We restrict to values of $n$ for which we can find an optimal solution (and run all of our approximation algorithms for many different choices of $b$) within a reasonable amount of time.

**Budget choices and runtime differences.** For each variant of MultiRepMFC, we used budgets $b$ ranging from 0 to $38t$ in increments of $2t$. This means that for the largest budget, we considered having an average of 39 representatives per component. For Fixed($\ell$)-MultiRepMFC, this corresponds to $\ell$ values from 1 to 39 in increments of 2.

For a fixed budget $b$, DP-MultiRepMFC tends to take longer than Greedy-MultiRepMFC, but for a somewhat surprising reason. Although the time to find representatives is slower for DP-MultiRepMFC, this constitutes only a small fraction of the total runtime and is not the primary reason why DP-MultiRepMFC is slower for a fixed $b$. By checking runtime for different steps, we found that the increase in runtime for DP-MultiRepMFC is due primarily to the total time it takes to compute the distances between representatives and points outside each representative's component, when running MultiRepMFC. For a fixed $R$, recall the number of distances computed by MultiRepMFC($R$) to find $\hat{w}$ is

$$\sum_{i=1}^{t} |R_i| \cdot (|\mathcal{X}| - |P_i|).$$

Although this is bounded above by $(b+1)n$, in practice this value will depend on which components the representatives are assigned to. In particular, if a representative is contained in a component $P_i$ that is very large, then $(|\mathcal{X}| - |P_i|)$ may be significantly smaller than the bound $|\mathcal{X}|$. We found that the number of distances computed by DP-MultiRepMFC tends be much larger than the number of distances computed by Greedy-MultiRepMFC. This suggests that Greedy-MultiRepMFC may have a greater tendency to place representatives in large components, while DP-MultiRepMFC finds ways to distribute representatives differently in a way that leads to better spanning trees but more distance computations.

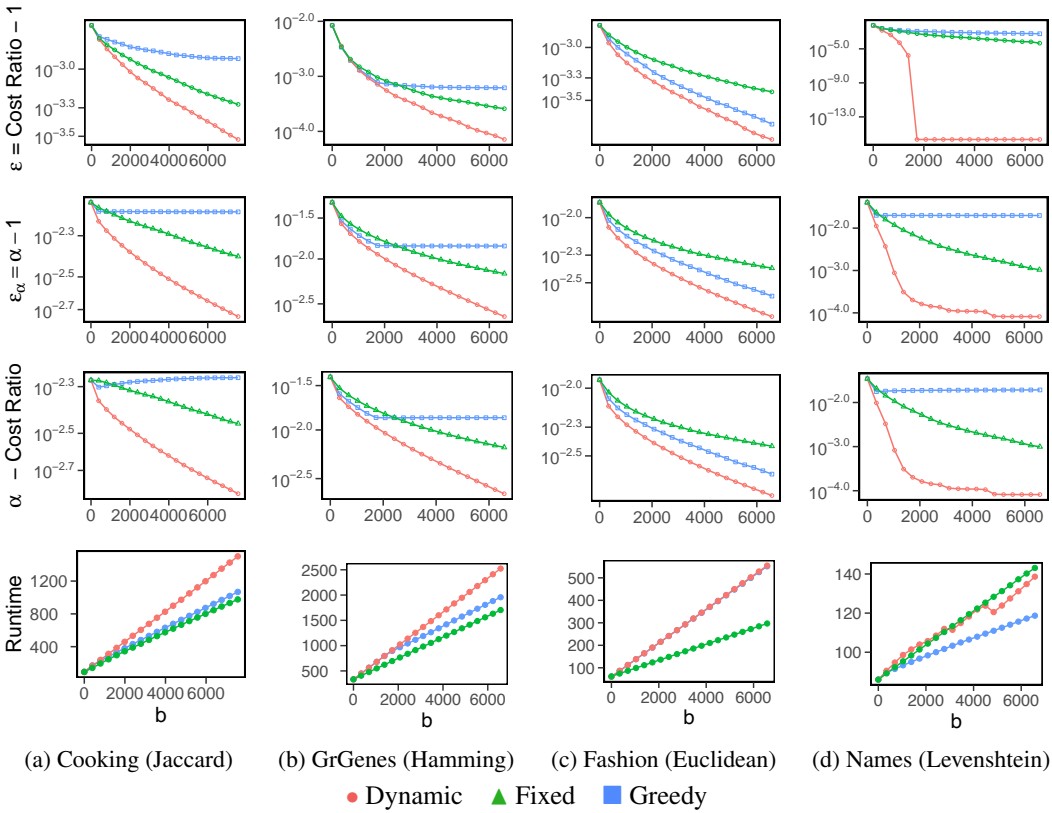

(a) Cooking (Jaccard)  (b) GrGenes (Hamming)  (c) Fashion (Euclidean)  (d) Names (Levenshtein)

• Dynamic   ▲ Fixed   ■ Greedy

Figure 4: We display the performance of each variant of MultiRepMFC as budget increases. Each point corresponds to running one method with a fixed budget $b$.

**Additional experimental results.** In Figure 4, we display results for *Cost Ratio* $-1$, $\alpha - 1$, and the gap between these two values as $b$ varies. These plots show the same basic trends as Figure 3, and the curves are more clearly aligned since there is exactly one point per value of $b$. We also display runtimes, as these differ for each method for each fixed choice of $b$.

Recall that the MFC *Cost Ratio* is defined by

$$\frac{w_{\mathcal{X}}(\hat{M}) + w_{\mathcal{X}}(E_t)}{w_{\mathcal{X}}(M^*) + w_{\mathcal{X}}(E_t)},$$

where $\hat{M}$ is the set of new edges added by MultiRepMFC, $M^*$ is the set of new edges added by an optimal solution to MFC, and $E_t$ is the initial forest. As another interesting point of comparison, Figure 5 displays results for the *Completion Ratio*, which only considers the weight of new edges, and does not incorporate the initial forest weight:

$$\frac{w_{\mathcal{X}}(\hat{M})}{w_{\mathcal{X}}(M^*)}.$$

We previously showed that it is not possible to design an algorithm with subquadratic complexity that comes with an a priori upper bound on this ratio in general metric spaces (Veldt et al., 2025). However, it is a useful quality measure to consider empirically, since it focuses more directly on the new edges found by the algorithm (recall that the weight of the initial forest is constant for every feasible solution for MFC). Figure 5 shows that adding even a small number of extra representatives improves the *Completion Ratio* even more significantly than the improvement to the MFC *Cost Ratio*. This indicates that MultiRepMFC is generally able to find much smaller edges to connect components in the initial forest. This is a promising sign for downstream applications where it is especially important to find small weight edges to connect components. For example, a key motivating application is to cluster a dataset $\mathcal{X}$ by mining a dendrogram associated with a spanning

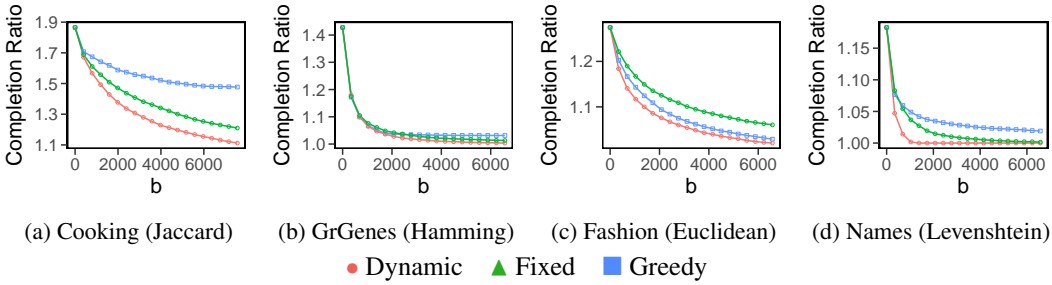

(a) Cooking (Jaccard)  (b) GrGenes (Hamming)  (c) Fashion (Euclidean)  (d) Names (Levenshtein)

● Dynamic  ▲ Fixed  ■ Greedy

Figure 5: For each dataset, we display the *completion ratio*: the weight of *new* edges added by MultiRepMFC, divided by the weight of the new edges added by an optimal solution for MFC. Adding a small number of extra representatives leads to an even more dramatic improvement to the completion ratio than to the MFC *Cost ratio*.

tree for $G_{\mathcal{X}}$. In this context, the initial forest provides a crude initial clustering of the data that can be refined and improved if one is able to truly find small weight edges between these initial clusters. The comparison between $w_{\mathcal{X}}(\hat{M})$ and $w_{\mathcal{X}}(M^*)$ is then especially important, even if the weight of the initial forest $w_{\mathcal{X}}(E_t)$ is large.

Finally, in Table 1 we provide a closer look at the various performance metrics ($\varepsilon$, $\varepsilon_{\alpha}$, number of distance calls, *Completion Ratio*, and runtime) for three choices of $b$, on all datasets.

Table 1: We give exact performance values for each variant of MultiRepMFC for three different values of $b$. Comp is the completion ratio, DC is millions of distance calls made by the entire pipeline, and RT is the runtime of the entire pipeline in seconds.

| Dataset | Alg | $b = 0$ | | | | | $b = t$ | | | | | $b = 2t$ | | | | |
|---|---|---|---|---|---|---|---|---|---|---|---|---|---|---|---|---|
| | | $\varepsilon$ | $\varepsilon_\alpha$ | Comp | DC | RT | $\varepsilon$ | $\varepsilon_\alpha$ | Comp | DC | RT | $\varepsilon$ | $\varepsilon_\alpha$ | Comp | DC | RT |
| Cooking | DP | 0.0022 | 0.0075 | 1.85 | 58.0 | 99.8 | 0.0017 | 0.0060 | 1.67 | 104.3 | 174.0 | 0.0014 | 0.0054 | 1.56 | 150.6 | 248.6 |
| | Greedy | 0.0022 | 0.0075 | 1.85 | 58.0 | 99.6 | 0.0018 | 0.0068 | 1.71 | 102.9 | 179.3 | 0.0017 | 0.0068 | 1.68 | 133.9 | 236.6 |
| | Fixed | 0.0022 | 0.0075 | 1.85 | 58.0 | 100.0 | 0.0018 | 0.0071 | 1.69 | 88.4 | 150.5 | 0.0016 | 0.0068 | 1.62 | 118.7 | 206.0 |
| Fashion | DP | 0.0017 | 0.0134 | 1.29 | 45.5 | 56.3 | 0.0011 | 0.0086 | 1.19 | 75.1 | 82.4 | 0.0009 | 0.0071 | 1.15 | 104.8 | 108.7 |
| | Greedy | 0.0017 | 0.0134 | 1.29 | 45.5 | 56.3 | 0.0012 | 0.0097 | 1.21 | 75.1 | 82.4 | 0.0010 | 0.0084 | 1.18 | 104.8 | 108.6 |
| | Fixed | 0.0017 | 0.0134 | 1.29 | 45.5 | 56.4 | 0.0013 | 0.0108 | 1.23 | 64.8 | 69.0 | 0.0012 | 0.0094 | 1.20 | 84.1 | 82.1 |
| Names | DP | 0.0061 | 0.0408 | 1.20 | 415.9 | 88.4 | 0.0015 | 0.0116 | 1.05 | 437.2 | 92.7 | 0.0004 | 0.0038 | 1.01 | 458.6 | 96.6 |
| | Greedy | 0.0061 | 0.0408 | 1.20 | 415.9 | 88.4 | 0.0025 | 0.0209 | 1.08 | 432.9 | 91.7 | 0.0019 | 0.0209 | 1.06 | 443.8 | 93.5 |
| | Fixed | 0.0061 | 0.0408 | 1.20 | 415.9 | 88.4 | 0.0027 | 0.0241 | 1.09 | 426.9 | 91.2 | 0.0016 | 0.0163 | 1.05 | 437.8 | 94.0 |
| GG | DP | 0.0084 | 0.0455 | 1.44 | 61.9 | 270.1 | 0.0034 | 0.0257 | 1.18 | 91.0 | 388.2 | 0.0020 | 0.0196 | 1.10 | 120.2 | 506.0 |
| | Greedy | 0.0084 | 0.0455 | 1.44 | 61.9 | 270.2 | 0.0034 | 0.0280 | 1.18 | 91.0 | 388.3 | 0.0019 | 0.0221 | 1.10 | 120.1 | 506.1 |
| | Fixed | 0.0084 | 0.0455 | 1.44 | 61.9 | 270.5 | 0.0034 | 0.0318 | 1.18 | 80.7 | 346.1 | 0.0020 | 0.0257 | 1.10 | 99.5 | 421.3 |

