# OpenReview forum: "Better Learning-Augmented Spanning Tree Algorithms via Metric Forest Completion"
_ICLR.cc/2026/Conference — ICLR 2026 Poster_

### Official Review · Reviewer_wWKo · 2025-10-21

**Soundness:** 3
**Presentation:** 4
**Contribution:** 3
**Rating:** 6
**Confidence:** 3

**Summary:**

The submission improves upon the recent learning-augmented algorithm of Veldt, Stanley, Priest, Steil, Iwabuchi, Jayram and Sanders. It provides a tighter analysis which explains the previous gap between theoretical lower bounds and experimental performance, and also develops a related means of interpolating between slower approaches (with worst-case guarantees) and approximation.

**Strengths:**

The contributions are, as a whole, clearly articulated and nicely presented. The submission's contributions are relevant for Machine Learning research and provide a nice combination of theoretical results supported with auxiliary experiments. The results are non-trivial. Moreover, the submission is nicely written overall.

While not a strength per se, I found the openness and detail in which the authors discussed their use of LLMs refreshing.

**Weaknesses:**

The contributions could be seen as slightly incremental over the recent work of Veidt: the additional contributions "on top" of Veidt are not that novel/substantial (this applies in particular to Subsection 3.1). On the other hand, the motivation for the theoretical contributions of Subsection 3.2 and the experiments in Section 4 is weaker than for the improved bounds; these show that one can interpolate between the exact and approximate procedures by obtaining larger representative sets R (via a DP with approximation guarantees), which is not that surprising.

I view both of the above as "minor" since in both respects, the submission only seems to be slightly below what I would consider the average for a combined theory+experimental ICLR paper.

*Typos and Minor Suggestions*

-Top of page 6: use \qedhere to place qed in correct location.

-Page 2: "approximations factors" ->  "approximation factors"

-Page 2: "... a disjoint set of trees called the initial forest, such that each of the n points belongs to one component in the forest." This sentence is unclear, as it does not specify what are the vertices of the forest. The formalization on page 3 is clear, however.

-Page 6: "This methods" -> "This method"

**Questions:**

On page 1, the submission attempts to provide a more formal footing for learning-augmented algorithms as follows:

"The goal (of learning-augmented algorithms) is to design an algorithm that is consistent, meaning that it produces near-optimal outputs when the prediction is good, and robust, meaning that it recovers the same worst-case guarantees as a prediction-free algorithm when the prediction is bad"

However, isn't it the case that *every* procedure which is "consistent" can be made "robust" by simply running the algorithm achieving worst-case running time in parallel? After all, worst-case guarantees are typically provided in O-notation (and this also holds for the lower bounds for MinTSP provided by (Indyk 1999)), so a factor-2 decrease in the running time is still tight w.r.t. the worst-case lower bound. If the above is indeed a gap, please suggest an alternative formulation of the goal of learning-augmented algorithms - would it simply be achieving the "consistency" property?

Additionally, the description of the purpose of the experiments in the Introduction was a bit unclear to me. Based on my current understanding, they demonstrate that the "novel" DP you develop towards achieving your theoretical guarantees also outperforms greedy/trivial solutions in practice, but feel free to comment on (and potentially correct) this.

---

> ### Author Response · Authors · 2025-11-20
> **Response to review**
>
> Thanks for your review. We appreciate you pointing out the typos and minor changes to be corrected. We have updated these in a new version of the manuscript (see updated pdf).
>
> Regarding your question about notions of robustness and consistency: this idea of making a consistent algorithm robust by running a prediction-free algorithm does indeed come up a lot in discussions of learning-augmented algorithms. There are a few reasons why we still find it helpful to present learning-augmented algorithms as we have done in the introduction:
>
> ●	In many settings, running a prediction-free algorithm in parallel can in theory make a consistent algorithm robust. However, there are a wide variety of improvements that are targeted using learning-augmented algorithms (e.g., runtimes, query complexities, approximation factors, competitive ratios in online algorithms, etc.). We’d prefer to be cautious and not make a claim that every consistent algorithm can be made robust by running a parallel prediction-free algorithm, since the exact notion of consistent and robust can depend on the context.
>
> ●	In this part of the introduction we are mainly attempting to summarize the broad literature on learning-augmented algorithms, and the terminology we have used directly follows the standard terminology in the literature. While a discussion about exactly when and how consistent algorithms can be made robust (and whether or not this is always possible) could be interesting, we are not sure our introduction is the right place for this discussion.
>
> ●	Even when a consistent algorithm is---in an asymptotic theoretical sense---effectively the same as a consistent and robust algorithm, there are still practical considerations that make it worthwhile to explicitly consider notions of both consistency and robustness as presented. We’d like to highlight that our intent in the introduction is not solely to provide theoretical foundations for learning-augmented algorithms, but also provide good high-level intuition for this framework, since both theory and practice matter for learning-augmented algorithms.
>
> > "Additionally, the description of the purpose of the experiments in the Introduction was a bit unclear to me. Based on my current understanding, they demonstrate that the "novel" DP you develop towards achieving your theoretical guarantees also outperforms greedy/trivial solutions in practice, but feel free to comment on (and potentially correct) this."
>
> Thanks for pointing out the confusion. We wonder if perhaps there is some confusion simply because our commentary in the introduction about experiments directly follows our commentary about the DP algorithm. Note that the commentary following the sentence “As a final contribution…” is meant to refer to what the numerical experiments tell us about all of our results and algorithm variants, and not just what these experiments tell us about the comparison between the DP variant and the other variants.
> To fix this potential confusion, we have added a paragraph break at the sentence "As a final contribution..." If there was something else beyond this causing confusion, please let us know and we are happy to follow up again.
>
> Thanks again for your review. If you have any further clarification or questions, we are happy to respond.

---

> > ### Comment · Reviewer_wWKo · 2025-11-24
> >
> > Hello,
> >
> > I find it strange to insist on keeping the following formulation on page 1:
> >
> > "The goal is to design an algorithm that is consistent, meaning that it produces near-optimal outputs when the prediction is good, and robust, meaning that it recovers the same worst-case guarantees as a prediction-free algorithm when the prediction is bad."
> >
> > when there are doubts regarding the soundness of the formulation of the goals of the whole research direction. Can every algorithm be made robust by running a worst-case algorithm in parallel? At least in the standard computational models considered in complexity theory, this seems to be true. Your response is very vague and does not help clarify this point.
> >
> > To be clear: the submission has concrete contributions that are detached from the claim quoted above. My suggestion is to simply rephrase that paragraph in a way which avoids the spurious description of learning-augmented algorithms. (I do not insist on having an answer to the previous question as long as the sentence is gone or suitably altered.)

---

> > > ### Author Response · Authors · 2025-12-02
> > >
> > > Hi,
> > >
> > > To avoid possible confusion regarding the nuances in this terminology, we can change the line
> > >
> > > "The goal is to design an algorithm that is consistent, meaning that it produces near-optimal outputs when the prediction is good, and robust, meaning that it recovers the same worst-case guarantees as a prediction-free algorithm when the prediction is bad"
> > >
> > > to
> > >
> > > "The goal is to design an algorithm that can obtain better than worst-case guarantees using the prediction."
> > >
> > > Thanks again for your feedback and review of our work.

---

### Official Review · Reviewer_VpQA · 2025-10-29

**Soundness:** 3
**Presentation:** 3
**Contribution:** 3
**Rating:** 6
**Confidence:** 4

**Summary:**

This submission deals with the problem of computing an approximate minimum spanning tree (MST) in subquadratic time when a partial solution is already given. More precisely, we are given a metric instance of the MST problem and an initial forest. The goal is to compute approximately a minimum spanning tree in subquadratic time among all spanning trees that contain the given forest. This problem is known as the metric forest completion (MFC) problem and it is motivated in the context of learning-augmented algorithms in which, e.g., a learning algorithm provides already a partial prediction for the minimum spanning tree.

The submission is heavily based on the research by Veldt et al. (2025), who first considered the MFC problem. The quality of the initial forest is measured by a parameter γ, which corresponds to the overlap of the given initial forest with some MST and is defined as the ratio between the current weight of the given forest and an optimal choice of an optimal minimum spanning tree, where we only count the weight of edges if the corresponding endpoints are in the same component as in the given forest (γ=1 means an optimal overlap with some MST). Veldt et al. gave a 2.62-approximation for MFC and showed that if the initial forest has a γ-overlap then the final spanning tree is a (2γ+1)-approximation of an MST. They also conducted multiple experiments and concluded that even simple heuristics for the initial forest provide very good results on synthetic and real datasets.

As first main contribution, the paper at hand improves on the theoretical bounds by showing that the algorithm by Veldt et al. (2025) even achieves a 2-approximation for the MFC problem and a 2γ-approximation for the metric MST problem. Additionally, it is shown that these bounds are tight in the worst case by designing a family of instances with γ=1, where for a specific choice of representatives in each component of the initial forest the ratio between the MST and the optimal corresponding MFC solution approaches 2 in the limit. The second main contribution is a modification of the algorithm by Veldt et al. (2025). While the latter is based on choosing a single representative for each component of the initial forest, the generalized version considered in this submission, is allowed to choose multiple representatives in each component, where the total number of additional representatives is upper bounded by some value b. For this setting (called best representatives problem) the authors design a 2-approximation using a 2-approximation algorithm for k-center and dynamic programming. Additionally, they test their algorithms on some of the real-world instances used by Veldt et al. and show experimentally that slightly increasing the number of allowed representatives gives some improvement in solution quality with only a small increase in runtime.

The paper is generally well written and follows closely the notation and experimental setup of Veldt et al. The theoretical results are clearly written. The proof of the 2-approximation is similar to the proof ideas by Veldt et al., but uses a different technique to bound the final cost and show the corresponding approximation factors. Allowing more representatives in each component is a natural extension of the algorithm. The experimental results show a slight improvement in solution cost when using more representatives per component or allowing for a larger additional time budget beyond the initial case with having a single representative.

**Strengths:**

The authors improve the analysis of the existing approximation algorithm for the MCF problem and show tight bounds. The theoretical analysis is sound and the writing is good. The modified algorithm using additional representatives performs well in experiments.

**Weaknesses:**

The theoretical considerations rely very much on the work of Veldt et al. (2025) and neither the problem nor the algorithm is new. The extension towards additional representatives is quite canonical and does not provide better theoretical bounds. I have the impression that the results are slightly overstated: More representatives lead to better solutions but the difference seems to be rather small while the running time increases by a significant factor (see questions for more details).

**Questions:**

- The experimental section misses some additional information about the system on which the experiments were run.

- Additionally, the significance of including additional representatives with respect to the runtime is not clear. For example, on the Cooking dataset it seems that the standard algorithm (b=0) did run in the experiments in approximately 100 seconds while the additional variants for selecting additional representatives can take more than 1000 seconds while improving the approximation factor only slightly. The improvement looks more severe due to the scaling and truncation of the diagram but if I understand Figure 2 correctly then already the original algorithm achieves an approximation factor of around 1,01. A summary in form of a table and different values of b could be useful here.

- Also, in the experimental evaluation it is not exactly clear how the setup is precisely done, as the value b is not part of the description but only implicitly mentioned as being an effect of increasing values of b. I would have preferred an analysis as in the appendix, section E, where the value of b gradually increases, or even an additional plot with b on the x-axis and the corresponding runtime (increase) on the y-axis.

- In Figure 2, for some of the greedy plots we have at some point straight lines, though b increases. Though the explanation, written in the section regarding question 2 is also making sense, I am still surprised that no further decrease in cost is being observed, even if much more representatives are allowed and costs are greedily decreased.

---

> ### Author Response · Authors · 2025-11-20
> **Response to review**
>
> Thanks for your review. We respond to each concern and question in turn:
>
> > "The theoretical considerations rely very much on the work of Veldt et al. (2025) and neither the problem nor the algorithm is new. The extension towards additional representatives is quite canonical and does not provide better theoretical bounds."
>
> We do agree that choosing multiple representatives per cluster is a straightforward idea. However, the more important contribution of our work is not just to use this approach, but to prove concrete and non-trivial theoretical results for the strategy.
>
> We'd also like to point out that providing a tighter and better analysis for an existing algorithm is a common and often very valuable contribution even if the algorithm itself is not new.
>
> We’d add that Theorem 1 does provide a new and better theoretical bound for using multiple representatives, even if it is an instance-specific bound. While this is still 2 in the worst case, our empirical results demonstrate that this bound enables us to quickly certify in practice that we are getting near optimal solutions (far better than 2), without ever having to compute an optimal solution (which in practical applications is infeasible).
>
> > "The experimental section misses some additional information about the system on which the experiments were run."
>
> Thanks for pointing this out. We've updated details in the experimental setup portion of the appendix.
>
> > "Additionally, the significance of including additional representatives with respect to the runtime is not clear. For example, on the Cooking dataset it seems that the standard algorithm (b=0) did run in the experiments in approximately 100 seconds while the additional variants for selecting additional representatives can take more than 1000 seconds while improving the approximation factor only slightly. The improvement looks more severe due to the scaling and truncation of the diagram but if I understand Figure 2 correctly then already the original algorithm achieves an approximation factor of around 1,01."
>
> We agree that even for b = 0, the approximation ratio for the full spanning tree (which includes the weight of the initial forest) is quite good, but we still see improvements as b increases, and even for large b the runtime is generally much better than the slow optimal algorithm.
>
> We have also added some new plots in the appendix and accompanying commentary (see updated pdf) that demonstrate that the improvement is even more dramatic (even for small increases in runtime) when specifically considering the weight of *new* edges that are added to the initial forest. We expect this will be an important consideration in downstream applications (e.g., downstream clustering applications where the “smallness” of new edges added is important for refining an initial clustering into a better one).
>
> >  "A summary in form of a table and different values of b could be useful here."
>
> Thanks for the suggestion. We have added Table 1 in the appendix to more precisely show how our approximation ratios and other metrics change for the first few values of b, since these precise values are not easy to read directly from the figure."
>
> > "Also, in the experimental evaluation it is not exactly clear how the setup is precisely done...I would have preferred an analysis as in the appendix, section E, where the value of b gradually increases, or even an additional plot with b on the x-axis and the corresponding runtime (increase) on the y-axis."
>
> The plot in the appendix E (with b in the x-axis) can indeed also be helpful, but the concern is that this does not provide as direct of a comparison between methods, since allocating b representatives to (say) Greedy is incomparable to allocating b representatives to (say) Dynamic programming. In the main text, we have tried to provide as fair of a comparison as possible by showing quality vs. runtime, while trying to make this as clear as possible by stating in the caption that "Each point corresponds to running one method with a fixed budget b."
>
> In the appendix, we further specify the values of b that were used to produce plots. If space permits, we are happy to move that information from the appendix to the main text.
>
> We have also added another row of plots in the appendix with runtimes as b increases. Thanks for suggesting it.
>
> > "In Figure 2, for some of the greedy plots we have at some point straight lines, though b increases. Though the explanation, written in the section regarding question 2 is also making sense, I am still surprised that no further decrease in cost is being observed..."
>
> It is indeed interesting and a little surprising that the greedy plots plateau. We're glad you agree the explanation in the main text makes sense. Note that the curves are still decreasing, but only by a very small amount.
>
> Thanks again for your review. If you have any further clarification or questions, we are happy to respond.

---

> > ### Comment · Reviewer_VpQA · 2025-11-23
> >
> > Dear Authors,
> >
> > Thank you very much for your detailed answer and the additional information provided. While some details are now clearer to me, my overall impression has not changed and I will keep my current score.
> >
> > Best wishes,
> > Reviewer

---

> > > ### Author Response · Authors · 2025-12-02
> > > **Thanks**
> > >
> > > Thanks for your reply. We are glad to hear our response helped clear up some details for you.

---

### Official Review · Reviewer_BGHP · 2025-10-31

**Soundness:** 3
**Presentation:** 3
**Contribution:** 3
**Rating:** 6
**Confidence:** 2

**Summary:**

This paper considers a learning-augmented variant of the classical metric minimum spanning tree (MST) problem. The authors study a setting proposed by Veldt et al. (2025), where the algorithm has a-priori access to an initial forest with the goal to augment the initial forest to a spanning tree with the objective of minimising the total edge weight. This problem is called metric forest completion (MFC). Since both MST and MFC are known to require a running time of $\Omega(n^2)$, the goal is to find approximation algorithms with a lower running time. The initial forest can be interpreted as a prediction of an MST, making MFC a learning-augmented problem. If there exists an MST that contains the initial forest, then optimally augmenting the initial forest leads to the computation of an MST. In general, the authors define a function $\gamma$ which measures "how close" the initial forest is to being augmentable into an MST.

As a main result, the authors give a $2$-approximation for a MFC, which implies a $2\gamma$-approximation for MST, with running time $o(n^2)$ if the number of components in the initial forest is $o(n)$. The algorithm is a generalisation of the algorithm by Veldt et al., which arbitrarily choses one representative from each component of the initial forest and finds the optimal augmentation that only uses edges incident to at least one representative. If the number of components is $o(n)$, then this algorithm yields a running time $o(n^2)$. The algorithm in this paper is based on the same idea but allows multiple representatives per component within a certain budget. As a first main result, the authors show that the algorithm is a $2$-approximation for a MFC and a $2\gamma$-approximation for MST, for a fixed arbitrary set of representatives.  This improves upon the guarantees of $2.62$ and $2\gamma+1$ by Veldt et al. As the second main result, the authors give a $2$-approximation for the problem of selecting the optimal set of representatives.This approximation first computes $b+1$ (the overall budget for representatives) potential representatives for each component by greedily solving $k$-center clustering instances. Then, it uses a knapsack DP to allocate the budget to the components. Finally, the authors give empirical experiments to compare the performance of the algorithm for different representative budgets and different representative selection methods.

**Strengths:**

* The proof of Theorem 1 is very clear and simple, and at the same time improves the guarantees by Veldt et al. In my opinion, this is a very nice contribution.
* The empirical experiments nicely illustrate how the choices (budget, selection of representatives) in the algorithmic framework impact performance of the algorithm.
* The paper studies a well-motivated problem. The framework of learning-augmented algorithms is still mainly studied for online problems: According to the online repository, there are more than 170 paper studying online problems and only 30 studying running time improvements. Further work on learning-augmented settings to improve running times is an important contribution.

**Weaknesses:**

* The impact of allowing more representatives seems to be mainly studied empirically. It would be nice to have a discussion of theoretical results depending on the budget.
* The new analysis of the algorithm does not require many new technical ideas.
* For the sake of transparency, it would be nice to already mention in the introduction that the running time depends on the number of components.

**Questions:**

* Do you think there is hope for theoretical results that quantify the impact of allowing a larger budget for representatives?

---

> ### Author Response · Authors · 2025-11-20
> **Response to review**
>
> Thanks for your review. We respond to each question in turn:
>
> >"The impact of allowing more representatives seems to be mainly studied empirically. It would be nice to have a discussion of theoretical results depending on the budget."
>
> >"Do you think there is hope for theoretical results that quantify the impact of allowing a larger budget for representatives?"
>
> Since the size of the set R in Theorem 1 depends directly on the budget b, we would actually describe Theorem 1 as already being a theoretical result that depends on the budget. However, if we understand correctly, you may just be asking whether we could prove approximation results where the budget b shows up explicitly in the approximation ratio.
>
> If so, we have made progress (by way of negative results) on this question by updating Theorem 3 to show that the 2-approximation is tight even for an arbitrary number of representatives, when those representatives are chosen arbitrarily (see updated pdf).
>
> This updated hardness result partially answers your question about the hope for theoretical results when allowing a larger budget. In particular, it proves that we cannot hope to improve on the worst-case 2 approximation, or obtain strictly improving approximations in terms of b, without making more assumptions.
>
> A natural follow-up question is whether we could get better guarantees in terms of b if we assume a specific strategy for choosing representatives. However, even this is likely to be very challenging (and may be impossible), since the approximation bound will depend heavily on the metric being used and the structure of the initial forest (which the MFC framework makes basically no assumptions about). This is an interesting question to consider, but fully resolving it is somewhat beyond the scope of this submission. Nevertheless, our empirical results confirm there is great value in having access to the instance-specific approximation ratio in Theorem 1, which in practice is very good and much better than what we would get from worst case pathological examples.
>
> >"For the sake of transparency, it would be nice to already mention in the introduction that the running time depends on the number of components."
>
> This is a good point, and we agree. We updated the intro to make this clear earlier on (see updated pdf)
>
> Thanks again for your review. If you have any further clarification or questions, we are happy to respond.

---

> > ### Comment · Reviewer_BGHP · 2025-11-26
> >
> > Dear Authors,
> >
> > Thank you for the response, this answers all my questions. My overall impression stays the same, and I will keep my score.
> >
> > Best regards,
> > Reviewer

---

> > > ### Author Response · Authors · 2025-12-02
> > > **Thanks**
> > >
> > > Thanks for the reply. We are glad to hear our response answered all your questions.

---

### Official Review · Reviewer_sJB9 · 2025-10-31

**Soundness:** 3
**Presentation:** 3
**Contribution:** 3
**Rating:** 8
**Confidence:** 4

**Summary:**

The paper considers MST with far fewer than $n^2$ distance queries via the Metric Forest Completion framework, generalizing the one-representative scheme to MultiRepMFC (multiple reps per component) and considering only edges touching reps; it proves an instance-specific bound $\alpha=1+\mathrm{cost}(P,R)/w(E_t)$, yielding an $\alpha$-approx for MFC and $\alpha\cdot \gamma$-approx for MST, which in particular improves the classic one-rep worst case from 2.62 to 2 for MFC and $(2\gamma+1)$ to $2\gamma$ for MST with a matching tight example; representatives are chosen via a shared-budget multi-instance $k$-center with a 2-approx.

**Strengths:**

1. Firstly, I appreciate that the paper is well-written and well-organized; I enjoy reading it. The authors provide sufficient intuition for each main theorem, making it easy to understand the main idea behind the improved algorithm.

2. This paper considers an interesting problem and uses a simple idea to get an improved algorithm. The algorithm is easy to implement, and thus, I expect it to have a positive impact in practice.

3. Casting representative selection as a shared-budget multi-instance k-center and giving a 2-approx (Gonzalez curves + DP allocation) is neat. This algorithm is simple but looks interesting to me, and it may be helpful for other related problems.

**Weaknesses:**

1. From the theoretical view, this work looks incremental. The idea of increasing the size of the representative set is standard and appears in many other classical problems (e.g., k-means++ and greedy k-means++). This is a weakness from a purely technical angle, but borrowing techniques from one problem to create a better approximation algorithm sounds like a good approach to me, especially for a non-theory conference.

2. Computing this larger representative set looks time-expensive to me, but the final running time is still in subquadratic time. Since several subquadratic Euclidean MST methods exist, a brief comparison explaining when your algorithm, MultiRepMFC, is preferable would help situate the work. I think it would be better to have a small table to compare distance query counts per algorithm variant or compare their precise running times.

**Questions:**

Are there conditional lower bounds (e.g., SETH-style or decision-tree/query-complexity) for achieving <2 for MFC with $o(n^2)$ queries?

---

> ### Author Response · Authors · 2025-11-20
> **Response to review**
>
> Thanks for your review. We respond to each comment in turn.
>
> > "The idea of increasing the size of the representative set is standard and appears in many other classical problems (e.g., k-means++ and greedy k-means++). This is a weakness from a purely technical angle, but borrowing techniques from one problem to create a better approximation algorithm sounds like a good approach to me, especially for a non-theory conference."
>
> We do agree that choosing multiple representatives per cluster is a standard idea, but our main contribution is not that we suggest this approach, but rather that we prove concrete and non-trivial theoretical results that hold for this approach.
>
> We also certainly agree that borrowing techniques from one problem is a good approach if it leads to better results. Indeed, we believe that in many ways it is actually a benefit that we are able to obtain these noticeable improvements without having to use a more complicated technical approach.
>
> > "Computing this larger representative set looks time-expensive to me, but the final running time is still in subquadratic time. Since several subquadratic Euclidean MST methods exist, a brief comparison explaining when your algorithm, MultiRepMFC, is preferable would help situate the work."
>
> We want to emphasize that the main focus of the MFC framework is that it applies to arbitrary metric spaces, and hence it is not directly comparable to EMST methods, which heavily rely on the assumption that the data is Euclidean. Nevertheless, MultiRepMFC is simple to implement and is a reasonable approach to use even on Euclidean data, so we have added some additional commentary in the appendix (see updated manuscript) to compare the subquadratic runtimes of MultiRepMFC against the types of subquadratic runtimes that are achievable for the much more restricting EMST setting.
>
> > "I think it would be better to have a small table to compare distance query counts per algorithm variant or compare their precise running times."
>
> Thanks for the suggestion. We have added several new plots to the appendix to display our numerical results from more perspectives. In particular, we have taken your advice and compared precise running times for the algorithm variants as b increases. On the last page of the updated manuscript, we included a table that lists running times, approximations achieved, distance query counts for a few values of b.
>
> > “Are there conditional lower bounds (e.g., SETH-style or decision-tree/query-complexity) for achieving <2 for MFC with o(n^2) queries?”
>
> This is an interesting question, and indeed one we have highlighted in Section 5 as a good open direction. We remark that we do not think a SETH-style bound would end up being the right approach, since SETH-style bounds deal with whether certain problems permit subexponential algorithms. Since we are focused on subquadratic algorithms for a problem that is easily solved in quadratic time, query-complexity lower bounds seem the more promising approach, but this seems to be a challenging question best left for future work.
>
> Thanks again for your review. If you have any further clarification or questions, we are happy to respond.

---

> > ### Comment · Reviewer_sJB9 · 2025-11-26
> > **Response**
> >
> > Thank the authors for addressing my questions. From a theoretical perspective, I still consider this work incremental. However, I don’t see this as a significant drawback for a non-theory conference, so I will maintain my score.

---

> > > ### Author Response · Authors · 2025-12-02
> > > **Thanks**
> > >
> > > Thanks for the follow-up, and once again for your review of the work.

---

### Official Review · Reviewer_YEtK · 2025-11-01

**Soundness:** 3
**Presentation:** 3
**Contribution:** 2
**Rating:** 4
**Confidence:** 4

**Summary:**

The paper considers the learning augmented minimum spanning tree (MST) problem for points in metric space. The authors propose solving this using the metric forest completion problem (MFC), where given an input of forest, goal is to find edges to form a full spanning tree, where the forest can be considered to be provided by an oracle (the learning augmented model). Previous work shows $\Omega(n^2)$ runtime is necessary for optimally solving MFC, and gives a 2.62 approximate algorithm. This works extends the previous algorithm and obtains 2 approximation for MFC and $(2\gamma + 1)$-approximation for learning augmented MST, where $\gamma$ measures how good the input forest is w.r.t optimal MST.

**Strengths:**

The previous work solved the MFC problem by considering one representative point per partition (in the forest), and then finding the set of edges minimizing the cost. The extension in this work is to instead consider multiple points in each partition. The problem of finding the best such set of points across different partitions is formulated as a variant of the $k-$center problem. The authors then show an extension of the classical 2 approximation $k-$center algorithm for this problem. This in turn, helps improve the approximation ratio to 2. Experiments show a few additional representatives help obtain much better solutions.

**Weaknesses:**

The idea of using multiple representatives per partition is a natural one. The analysis is clean but the algorithmic contributions are incremental. The authors argue for tightness of their result, but in the hard instance, the number of representative points per partition is 1.

**Questions:**

Can the authors provide hard instance for choosing multiple points arbitrarily per partition and the best possible approximation ratio for this?

---

> ### Author Response · Authors · 2025-11-20
> **Author response**
>
> Thanks for your review. We’d like to first make a couple of clarifications just to make sure the results of our paper are clear, since there are a couple statements in the review that summarize our contributions in a way that is not quite accurate.
>
> >“This works extends the previous algorithm and obtains 2 approximation for MFC and a $(2\gamma+1)$-approximation for learning augmented MST…”
>
> Our new approximation for MST is actually $2\gamma$. The $(2\gamma + 1)$-approximation was shown previously by Veldt et al. We also want to ensure it’s clear that the improved approximation ratios (2 for MFC and $2\gamma$ for MST)  hold for the existing algorithm of Veldt et al---one of our contributions is to provide a simpler and tighter analysis of the existing algorithm. Our generalized multirepresentative variant then leads further to the improved instance-specific results in Theorem 1.
>
> >“The authors then show an extension of the classical 2 approximation center algorithm for this problem. This in turn, helps improve the approximation ratio to 2.”
>
> To be clear, we use the classical 2-approximate k-center algorithm to design a 2-approximation for the BestReps problem, which is a separate problem from MFC. In this part of the paper, we are not actually “improving the approximation ratio to 2” per se, since there is no prior work on BestReps (hence nothing to “improve”). It is true that we “improve the approximation ratio to 2” for MFC (beating the previous 2.62 approximation), but we want to make sure it is clear that the latter improvement is not a result of the multi-representative generalization. The improved 2-approximation for MFC applies even to the previous algorithm of Veldt et al., and is independent from the 2-approximation for BestReps.
>
> > “The idea of using multiple representatives per partition is a natural one. The analysis is clean but the algorithmic contributions are incremental.”
>
> We agree that choosing multiple representatives is natural. We’d just highlight that the more important contribution of the paper is not simply the idea of choosing multiple representatives, but rather the concrete theoretical results we prove for this strategy.
> We hope also that the clarifications we made above help address concerns about the algorithmic contributions of this paper relative to prior work. In particular, our work extends previous research in multiple ways, including (1) improving the analysis of Veldt et al. to make it simpler while improving approximation ratios, (2) providing new instance specific approximations (Theorem 1) for a generalized algorithm that we show are very strong in practice, and (3) introducing the BestReps problem and showing a 2-approximation for it.
>
>
> > “The authors argue for tightness of their result, but in the hard instance, the number of representative points per partition is 1.
> >Can the authors provide hard instance for choosing multiple points arbitrarily per partition and the best possible approximation ratio for this?”
>
> To clarify, the tightness result in Theorem 3 was intended originally to show that the 2-approximation is tight for the existing algorithm of Veldt et al. (called MFC-Approx), which only considers 1 representative per component.
>
> That said, we are happy to report that yes, we can provide a hard instance for choosing multiple points arbitrarily per cluster. We have uploaded a new version of the manuscript that contains the updated proof, which generalizes the hard instance we originally used. We have also included a new figure for illustration. This updated result shows that Theorem 1 is tight if we have a pathological construction and we allow arbitrary representatives. This further motivates the research in Section 3.2 on choosing representatives strategically. It is worth noting that all the strategies we use for choosing representatives would fix the pathological choice of representatives in our updated Theorem 3.
>
> We hope these updates and clarifications help address your concerns. We are happy to respond if there are any follow-up questions.

---

> > ### Comment · Reviewer_YEtK · 2025-11-26
> > **Response to authors**
> >
> > I thank the authors for their reply and clarifications.
> > The $+1$ term was indeed a typo on my part, which I will correct in the text. Also, the generalization of $k$-center helps explore instance-specific bounds, which were not clearly stated in my original text. The new Theorem 3 also addresses my question about the hard instance. While I still view some aspects of this work as incremental, the rest of the contributions, particularly with respect to instance-specific bounds and runtime tradeoffs, are concrete. I would be happy to increase my score.

---

> > > ### Author Response · Authors · 2025-12-02
> > > **Thanks**
> > >
> > > Thanks for the follow-up! While we know that no further reviewer comments or updates will be possible, we very are glad to hear that our clarifications and reply were helpful, and that you were willing to increase your score.

---

### Meta-Review · Area_Chair_LrTY · 2026-01-05

**Summary:**

1. The algorithmic contributions are incremental
2. It lacks some details for empirical settings, e.g., runtime and explanation of the b value

**Reviewer Concerns:**

Most of the concerns have been addressed by the rebuttal.

**Reviewer Scores:**

Reviewer YEtK will increase the score from 4 to 6

---

### Decision · Program_Chairs · 2026-01-26

Accept (Poster)